# Weight Updates as Activation Shifts: A Principled Framework for Steering

Dyah Adila [1][*]   John Cooper [1][*]   Alexander Yun [1]   Avi Trost [1]   Frederic Sala [1]

## Abstract

Activation steering promises to be an extremely parameter-efficient form of adaptation, but its effectiveness depends on critical design choices—such as intervention location and parameterization—that currently rely on empirical heuristics rather than a principled foundation. We establish a first-order equivalence between activation-space interventions and weight-space updates, deriving the conditions under which activation steering can replicate fine-tuning behavior. This equivalence yields a **principled framework for steering design** and identifies the post-block output as a theoretically-backed and highly expressive intervention site. We explain why certain intervention locations outperform others and show that weight updates and activation updates play distinct, complementary functional roles. This analysis motivates a new approach—**joint adaptation**—that trains in both spaces simultaneously. Our post-block steering achieves accuracy within $0.2\%$–$0.9\%$ of full-parameter tuning, on average across tasks and models, while training only $0.04\%$ of model parameters. It consistently outperforms prior activation steering methods such as ReFT and PEFT approaches including LoRA, while using significantly fewer parameters. Finally, we show that joint adaptation often surpasses the performance ceilings of weight and activation updates in isolation, introducing a new paradigm for efficient model adaptation.[1]

## 1. Introduction

The massive parameter counts of modern large language models (LLMs) have spurred the development of parameter-efficient fine-tuning (PEFT) methods. While these significantly reduce the number of trainable parameters, they still require updating and storing weight-space modifications. A complementary family of methods, *activation steering*, further reduces adaptation costs by intervening directly on intermediate activations during the forward pass. By shifting the *locus* of change from the weights to the activations, these methods bypass much of the memory overhead typically associated with weight-space updates (Fig. 1, left).

Despite their empirical success, activation steering methods remain largely heuristic-driven. Current research in this area primarily relies on exhaustive trial-and-error to determine critical design choices, such as optimal parameterization and intervention sites. For instance, JoLA (Lai et al., 2025) treats the intervention site as a learnable parameter and offers little insight into the underlying mechanics. Furthermore, most prominent methods evaluate specific intervention points in isolation, lacking a comparative framework to explain why certain loci outperform others. Consequently, the design of steering interventions remains a black box process lacking generalizable principles.

In this work, we move beyond these heuristics by establishing a formal equivalence mapping between weight-space updates and activation-space interventions. By treating traditional weight-based fine-tuning as the gold standard target behavior, ***our framework reveals the precise conditions under which activation steering can faithfully replicate weight-space dynamics***. Our analysis identifies the **post-block output**—where the skip-connection is added back to the MLP output—as a highly expressive intervention locus (Figure 1, right). Unlike prior methods that operate on isolated pathways, this site modulates each layer's full residual stream. The resulting steering approach ***achieves accuracy within $0.2\%$–$0.9\%$ of full-parameter fine-tuning (SFT), on average across tasks and models, while training only $0.04\%$ of model parameters***. We also consistently outperform existing steering methods like ReFT, as well as PEFT methods despite using $15\times$ less parameters.

Theoretically, we explain why specific intervention sites outperform others and verify that post-block updates can mimic post-MLP updates under simple geometric assumptions. Beyond identifying this principled and highly expressive locus, we quantify the different functions expressed

---

[1]Department of Computer Science, University of Wisconsin-Madison. Correspondence to: Dyah Adila < adila@wisc.edu>, John Cooper < jfcooper2@wisc.edu>.

*Proceedings of the $43^{rd}$ International Conference on Machine Learning*, Seoul, South Korea. PMLR 306, 2026. Copyright 2026 by the author(s).

[1]Code is available in the following link: https://github.com/SprocketLab/steerling.git

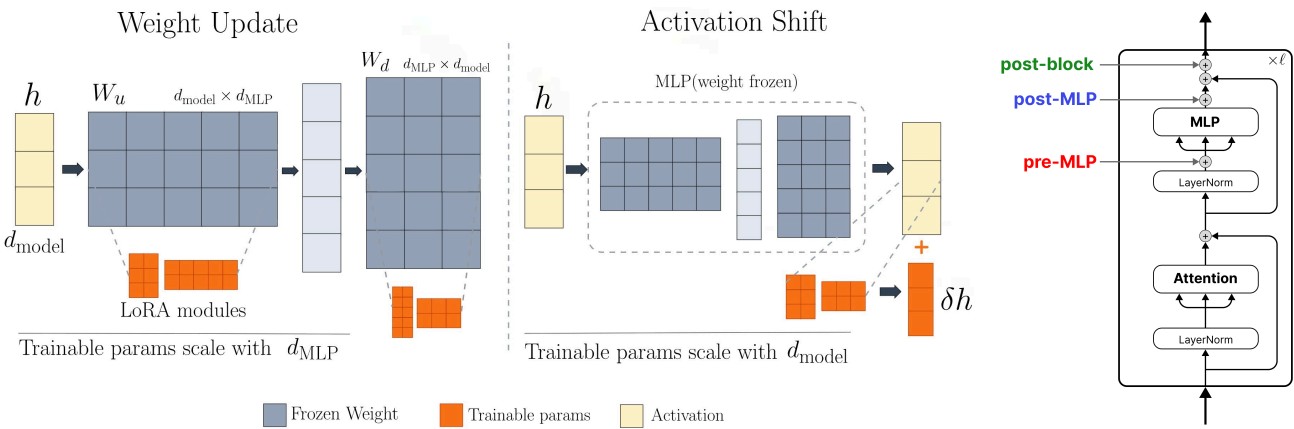

*Figure 1.* Left: Weight tuning scales with MLP dimension, while activation steering scales only with model dimension. Right: pre-MLP vs. post-MLP vs. post-block (ours). We steer after the skip connection is added back to the MLP output, accounting for both pathways.

by ***weight versus activation updates, revealing that they are fundamentally complementary***. These findings lead to us to explore a new approach: ***jointly learning in both weight and activation spaces***. This strategy surpasses the performance limits of either method alone by up to 3.8%; we achieve this by applying an orthogonality constraint that prevents the solution from collapsing into either individual subspace. We provide theoretical and empirical evidence that this constraint is crucial for maintaining independent learned solutions in each space.

**Our Contributions.** This work provides a principled foundation for activation-space adaptation, shifting the field away from empirical trial-and-error. Our contributions are:

- **First-Order Equivalence Framework:** We establish a formal mapping that identifies the conditions under which activation steering replicates weight-space fine-tuning.
- **Fine-tuning and Steering Separation:** While there are first-order similarities between these methods, we show they are fundamentally different when MLP feature maps behave differently than an identity map.
- **Identification of the Post-Block Locus:** Our framework identifies the post-block output as a principled and highly expressive intervention site. By integrating both the residual and MLP pathways, it enables steering to reach within 0.2%–0.9% of full-parameter fine-tuning on average across tasks and models, while training only 0.04% of parameters. This consistently outperforms interventions at isolated sub-layer sites.
- **Joint Adaptation:** We introduce a method for joint learning in weight and activation spaces. Using an orthogonality constraint to prevent functional redundancy, we enable a joint adaptation regime that surpasses the performance ceilings of either method in isolation by up to 3.8%.

## 2. Related Work

**Parameter Efficient Fine-tuning** Standard PEFT methods reduce adaptation costs by training small weight-space modules. This includes both approaches like classic bottleneck adapters (Houlsby et al., 2019), which insert trainable layers into the transformer block, and *reparameterization* approaches like LoRA (Hu et al., 2022) and its variants (DoRA (Liu et al., 2024), QLoRA (Dettmers et al., 2023)). While they significantly reduce fine-tuning costs, such weight-space modifications do not seek to use intermediate activations—the key component in steering.

**Activation Steering.** Activation steering intervenes on intermediate activations during forward pass, achieving parameter savings an order of magnitude greater than weight-based adapters. Initial research identifies latent subspaces for specific traits, such as truthfulness (Li et al., 2023a), style (Rimsky et al., 2024), function execution (Todd et al., 2023), or alignment (Adila et al., 2024), and steer the model towards the desired subspaces. More recently, methods like ReFT (Wu et al., 2024), LoFiT (Yin et al., 2024), and JoLA (Lai et al., 2025) have transitioned toward *trainable* interventions that minimize a standard fine-tuning loss.

However, these methods remain largely heuristic-driven; design choices regarding intervention locus and parameterization are typically determined through empirical search rather than mechanical understanding. While JoLA (Lai et al., 2025) attempts to unify these heuristics by treating design choices as hyperparameters to be optimized, this increases optimization complexity and does not change the black-box nature of steering design.

**Interplay between adaptation paradigms.** Recent research has sought to unify different adaptation paradigms through a mechanistic lens. Significant effort has been directed toward interpreting in-context learning (ICL) as a form of implicit weight fine-tuning (Dai et al., 2023;

Von Oswald et al., 2023; Zhang et al., 2024), with recent evidence suggesting these implicit updates are effectively rank-1 (Dherin et al., 2025). This provides a direct bridge to activation steering, as an activation update can be viewed as a rank-1 LoRA update constrained to a single direction. Furthering this connection, (Bigelow et al., 2025) established a link between ICL and steering activations toward pre-identified behavioral subspaces.

Our work provides the first analytical bridge between explicit weight-space fine-tuning and trainable activation adapters. By deriving the mathematical equivalence between these two spaces, we move beyond black-box heuristics to a principled understanding of optimal intervention loci. We go further, investigating joint learning across the weight and activation spaces, deriving methods for these updates to cooperate.

## 3. A Unifying Framework for Weight Updates and Activation Steering

We begin by describing both steering and fine-tuning within a multi-layer perceptron (MLP) submodule in a Transformer model and highlight their similarities. We begin by describing the similarities between steering and fine-tuning gradients in Section 3.2. Next, we establish an oracle useful for theoretical analyses as a clean learning target; using this, along with empirical evidence, shows that **post-block steering leads to expressivity not seen in post-MLP steering**. The locations of pre-MLP , post-MLP , and post-block steering can all be seen in Figure 1.

### 3.1. Setup and Notation

We use $\Delta\cdot$ for a small change to a variable, while $\delta\cdot$ is a learned update to a model parameter. Steering updates a model's activations by updating $h$ to $h \to h + \delta h$. The update $\delta h$, called an *adapter*, can depend on the input $h$. For example, in ReFT (Wu et al., 2024), $\delta h$ is a specific, low-rank linear function acting on $h$. An alternative is a low-rank autoencoder, $\delta h = W_2\phi(W_1 h)$, where $W_2 \in \mathbb{R}^{d \times r}$ and $W_1 \in \mathbb{R}^{r \times d}$. Fine-tuning is an update to the weights of a model, replacing $W$ with $W + \delta W$. The parameter $\delta W$ is a learned constant.

### 3.2. First Order Analysis

First, we study the relationship between steering and fine-tuning in order to extract insights into where steering interventions should take place. Consider the Gated Linear Unit (GLU), a common variant of MLP (Dauphin et al., 2017; Shazeer, 2020) used in most modern LLMs, such as Llama, Gemma, and Qwen. This module is parameterized as $\text{GLU}(h) = W_d(\phi(W_g h) \odot W_u h)$.

We compare two different small-norm perturbations to this function: a small change to the activations $h$ and a small change to the parameters $W_d, W_g, W_u$; with the form

$$\Delta\text{GLU}_{\text{steer}}(h) =$$
$$W_d\Big[\underbrace{(\phi'(a_g) \odot a_u) \odot (W_g\Delta h)}_{\text{gated path}} + \underbrace{\phi(a_g) \odot (W_u\Delta h)}_{\text{un-gated path}}\Big]$$
$$+ O(\|\Delta h\|^2),$$

and

$$\Delta\text{GLU}_{\text{FT}}(h) = (\Delta W_d)m +$$
$$W_d\Big[\underbrace{(\phi'(a_g) \odot a_u) \odot ((\Delta W_g)h)}_{\text{gated path}} + \underbrace{\phi(a_g) \odot ((\Delta W_u)h)}_{\text{un-gated path}}\Big]$$
$$+ O((\|\Delta W_d\| + \|\Delta W_g\| + \|\Delta W_u\|)^2).$$

where

$$a_g = W_g h, \qquad a_u = W_u h,$$
$$m = \phi(a_g) \odot a_u, \qquad y = W_d m,$$

This reveals a close relationship between steering and fine-tuning, differing only by the term $(\Delta W_d)m$.

How can we account for the $(\Delta W_d)m$ term not present in the steering perturbation? A simple way to accommodate it is by using a **post-MLP steering** intervention $\text{GLU}(h) \mapsto \text{GLU}(h) + \delta h$ rather than a pre-MLP steering intervention $\text{GLU}(h) \mapsto \text{GLU}(h + \delta h)$. Empirically, linear adapters are sufficient in our settings (Section 6.3), consistent with the first-order viewpoint.

> **Takeaway 1.** For an MLP with small perturbations to weights/activations, post-MLP steering can account for fine-tuning updates that pre-MLP steering cannot.

### 3.3. Post-Block and Post-MLP

Equipped with these insights, we study steering expressivity. To further analyze the relationship between steering and fine-tuning, we introduce a theoretical tool: the *oracle*. It is a freely-parameterized activation update that exactly matches the hidden state of a fully fine-tuned (SFT) model:

$$\delta h_{\text{oracle}} = h_{\text{FT}} - h_{\text{base}}.$$

While not a practical model, the oracle provides a clean learning target for understanding the expressivity of different steering locations. A key question is *where* this oracle should be applied. Placing it before or after the MLP captures only the MLP's contribution, missing effects from the attention sublayer and skip connection. Figure 2 confirms this: MLP outputs account for only 40–70% of total block output magnitude across layers. This motivates placing the

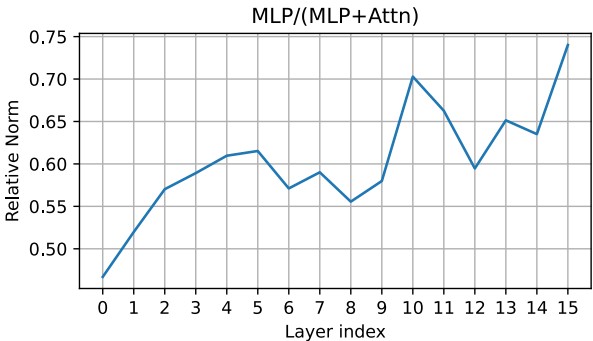

*Figure 2.* Average MLP output norm with respect to the layer's block output for Llama-3.2B on Winogrande. Post-MLP steering covers $\leq 70\%$ of the change from finetuning.

oracle *post-block* —after the skip connection—to capture the full residual stream update.

A key advantage of this oracle is that it enables layer-wise analysis: rather than reasoning about adapter interactions across the full model, we can study each layer independently. One might worry that learned adapters collaborate across layers in ways that violate this decomposition. However, if such an oracle—specifically, one that is close to a linear transformation of the input—can match the fine-tuned model at each layer, then steering the same type of adapters is *at least* as expressive as fine-tuning. Empirically, we find this to be the case: linear steering is sufficient to recover fine-tuned behavior (see Section 6). For our theoretical results, we typically have a vector-valued target learned under MSE loss. These vector-valued targets come from this oracle.

With the similarity between steering and fine-tuning understood, we investigate how closely a post-block steering can mimic a post-MLP steering. These will be matrices multiplying the associated component of the base model (either the output of just the MLP or the entire block).

**Theorem 3.1.** *Let $V$ and $V'$ be the right-singular matrices of $h + \mathrm{Attn}(h) + \mathrm{GLU}(h + \mathrm{Attn}(h))$ and $A_p\mathrm{GLU}(h + \mathrm{Attn}(h))$ respectively, and let $X, Y$ be defined as $Y_i^{mlp} := h_i + \mathrm{Attn}(h_i)$ and $Y_i^{pa} := \mathrm{GLU}(X_i)$. Then, the optimal relative error satisfies*

$$\min_A \frac{\|A(Y^{pa} + Y^{mlp}) - A_pY^{pa}\|_F^2}{\|A_pY^{pa}\|_F^2} = \sum_{i=1}^{d} \frac{\sigma_i^2}{S^2}\sin^2\theta_i,$$

*where $\sigma_i$ is the $i$-th singular value of $A_pY^{pa}$ and $\theta_i$ is the $i$-th principle angle between $V$ and $V'$ and $S^2 = \sum_{j=1}^{d}\sigma_j^2$.*

*Proof.* See Appendix C.1. □

This proposition tells us that the ability for a post-block steering to match a post-MLP steering depends on how similar the output of the GLU and the output of $h + \mathrm{Attn}(h)$ are. The greater the difference, the greater the principal angles (Stewart & Sun, 1990) will be, and therefore the greater the error. When the skip connection is relatively small, these two subspaces will be unlikely to differ much.

An unfortunate aspect of this error is that we are working with the *right* principal angles rather than the left ones. Principal angles measure the difference between two subspaces, where the $k$-th principal angle is the smallest angle between the two in directions orthogonal to the previous $k-1$ principal angles. The theorem above indicates the relevant quantity relating post-MLP steering $A_p$ to post-block steering $A$ is the similarity in the subspaces spanned by the kernel matrix of the post-MLP $X$ and the kernel matrix of the post-block activations $Y^{pa} + Y^{mlp}$. These are identical to the subspaces spanned by their right singular vectors $V$ and $V'$. It is important to note that this is measuring the kernel matrix rather than the covariance matrix, indicating a similarity needs to exist in sample-space rather than feature-space. What matters is that the geometry of the activations does not change much between before and after the MLP. If the MLP distorts the sample-to-sample geometry greatly, changing the kernel matrix, the resulting $\sin\theta_i$ will increase, decreasing the possible quality between post-MLP and post-block steering.

To gain intuition about how $A_pY^{pa}$ and $Y^{pa} + Y^{mlp}$ relate, note that $V$ and $V'$ diagonalize $(Y^{pa})^\top A_p^\top A_pY^{pa}$ and $(Y^{pa} + Y^{mlp})^\top(Y^{pa} + Y^{mlp})$, representing the relationships between the data-points. If $A_pY^{pa}$ and $^Ypa + Y^{mlp}$ keep similar points close and non-similar points distant, then $V$ and $V'$ will be similar, shrinking their principal angles.

As long as the MLP does not perturb the geometry of the data sufficiently, post-MLP steering can be approximately replicated by post-block steering. It thus makes sense to do the more expressive post-block steering instead.

> **Takeaway 2.** Post-block steering can more directly cover updates to attention from the skip-connection than post-MLP steering can.

## 4. Steering and Fine-Tuning Differences

Despite their similarities, we now provide insights into **important differences between steering and fine-tuning**. We investigate what takes place when we use *both* steering and fine-tuning simultaneously. In Section 4.1, we will see that the naive approach to this leads to both components learning similar subspaces—producing little improvement. However, this can be mitigated through orthogonality constraints between the two components (Section 5.2).

The first core difference between steering and fine-tuning is given by the following proposition. First, to get a handle

on the notation, we condense the MLP into $W_2 F(h)$. Here, $W_2$ is the down-projection for the MLP, $h$ is the input to the MLP, and $F$ encapsulates the rest of the MLP machinery (note that the parameterization of $F$ will be different for different choices of MLP, such as a GLU). The result after the residual connection is written as $\hat{g}(h) = h + W_2 F(h)$. Post-block steering will act as a matrix product on the left by $I + \delta h$ in the post-MLP (identity + low rank on the residual) and fine-tuning will replace $W_2$ with $W_2 + \delta W_2$.

**Proposition 4.1.** *Fix the feature map $F$ (thought of as the hidden state after non-linearity in an MLP) and let*

$$\hat{g}(h) = (I + \delta h)(h + (W_2 + \delta W)F(h))$$

*be the hybrid weight-update and post-block steering of $g_{\text{base}}(h) = h + W_2 F(h)$. Then, $\hat{g}_i$ can represent almost any linear combination of the inputs $\{h_j\}$ and the features $\{F_j\}$. Furthermore, if either $\delta h = 0$ or $\delta W = 0$, which correspond to fine-tuning and steering, respectively, then arbitrary linear combinations become impossible.*

Specifically, fine-tuning ($\delta h = 0$) forces $\hat{g}$ to keep the $h$ term fixed, and steering ($\delta W = 0$) forces $\hat{g}$ to join updates on $h$ and $F(h)$, leading to the same update on both. Without both, it is impossible to update $h$ and $F(h)$ separately.

> **Takeaway 3.** Steering and fine-tuning can express different functions. Together, they are *more* expressive.

**An intuitive example.** For a concrete, simple example, consider a task in $\mathbb{R}^2$ where both $h$ and $F(h)$ are constrained within the $h$-axis. The base model performs:

$$h \mapsto \begin{bmatrix} h + F(h), & 0 \end{bmatrix}^\top.$$

Suppose, for example, that $F(h) = \text{ReLU}(h)$.

- **Fine-tuning** only effects $F$. If $F$ is unable to match the identity function $F(h) = h$, then a linear fine-tuning of the output of $F$ is restricted to reweighting $F$:

$$h \mapsto \begin{bmatrix} h + \alpha_1 F(h), & \alpha_2 F(h) \end{bmatrix}^\top.$$

It is possible that tuning parameters within $F$ itself can modify the output of $F$, but again, if $F$ cannot represent the identity, the tuned model cannot use $h$ in their updates.
- **Steering** affects $h + F(h)$ as a unit. So, $h$ and $F(h)$ cannot be separated if they are already in the same subspace, as is the case in our example:

$$h \mapsto \begin{bmatrix} \beta_1(h + F(h)), & \beta_2(h + F(h)) \end{bmatrix}^\top.$$

Note that in practice, it is possible that $h$ and $F(h)$ are not in the same subspace and could be separated. This would, however, require matrices with large coefficients/norms to amplify these differences to a usable scale.

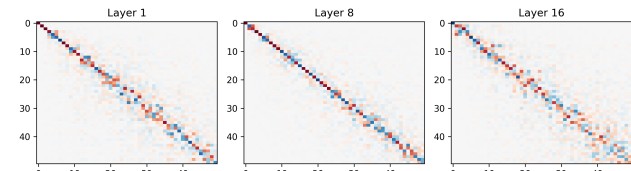

*Figure 3.* Joint training when done naively. The dot products between the top right singular vectors of the initial updates for both weight and activation adapters. Weight and activation adapters learn to fit the same subspace early on, represented by the diagonal.

- **Joint training** gives us both. Fine-tuning can split $F$ into the second dimension. Steering then can weight both terms separately, combining them into arbitrary linear combinations. This is even more important for a task limited to the first dimension:

$$h \mapsto \begin{bmatrix} \gamma_1 h + \gamma_2 F(h), & 0 \end{bmatrix}^\top.$$

The important part in this example is *disentangling the skip-connection*. Without additional dimensions to separate these effects, steering will still have these pieces tied together. Later in Section 6.4, we confirm that in real datasets, **joint training surpasses the performance ceiling of fine-tuning and steering alone**.

### 4.1. Joint Learning in Weight and Activation Space

Equipped with the insight that fine-tuning and steering play different functional roles and can be made more expressive when combined, we investigate **joint adaptation**—a strategy that learns in both spaces simultaneously.

Naively, we could train the parameters in both LoRA-style weight updates and the steering parameters. However, this turns out to work quite poorly, barely out-performing either weight-updates or steering on their own. This stems from the parameters learning in the same subspace.

**Proposition 4.2.** *When learning the simplified model*

$$y'(x) = (I + \delta h)(x + (W + \delta W)F(x))$$

*for learned matrices $\delta W$ and $\delta h$, early in training under MSE loss between $y$ and $y'$, these matrices have their dominant singular vectors coming from the top singular vectors of $RY^\top$ and $RF^\top$, where $R$, $F$, and $Y$ are matrices comprising $y - y'(x)$, $F(x)$, and $x + WF(x)$. If these two coincide, the two matrices learn the same subspace.*

*Proof (sketch).* When $\delta h, \delta W \approx 0$, $y'$ can is approximately

$$y'(x) \approx x + WF(x) + (\delta h)(x + WF(x)) + (\delta W)F(x),$$

or, with the above matrix notation,

$$Y'(X) \approx Y + (\delta h)Y + (\delta W)F.$$

Table 1. Comparison against SFT. Δ is performance difference relative to SFT. LoRA, ReFT and Ours all use rank 8

| Model | Method | Params (%) | BoolQ | Δ | WinoG | Δ | ARC-C | Δ | GSM8K | Δ | AQuA | Δ | ListOps | Δ | Avg. |
|---|---|---|---|---|---|---|---|---|---|---|---|---|---|---|---|
| Llama-3.2-1B | SFT | 100% | 88.2 | 0.0 | 58.2 | 0.0 | 60.3 | 0.0 | 32.2 | 0.0 | 36.2 | 0.0 | 66.2 | 0.0 | 0.0 |
| | LoRA | 0.45% | 84.7 | -3.5 | 57.0 | -1.2 | 61.6 | +1.3 | 31.8 | -0.4 | 33.6 | -2.9 | 60.0 | -6.2 | -2.2 |
| | ReFT | 0.04% | 84.1 | -4.1 | 49.3 | -8.9 | 57.0 | -2.7 | 31.6 | -0.6 | 30.2 | -6.0 | 49.3 | -16.9 | -6.5 |
| | Ours | 0.04% | 86.2 | -2.0 | 60.1 | +1.9 | 60.3 | 0.0 | 31.5 | -0.7 | 36.5 | +0.3 | 65.4 | -0.8 | -0.2 |
| gemma-3-1b | SFT | 100% | 83.3 | 0.0 | 51.4 | 0.0 | 48.2 | 0.0 | 23.4 | 0.0 | 32.7 | 0.0 | 65.8 | 0.0 | 0.0 |
| | LoRA | 0.45% | 84.7 | +1.4 | 51.6 | +0.2 | 50.6 | +2.4 | 22.6 | -0.8 | 31.6 | -1.1 | 53.6 | -12.2 | -1.7 |
| | ReFT | 0.04% | 76.2 | -7.1 | 49.2 | -2.2 | 27.8 | -20.4 | 11.6 | -11.8 | 24.6 | -8.1 | 50.3 | -15.5 | -10.9 |
| | Ours | 0.04% | 82.0 | -1.3 | 50.8 | -0.6 | 48.9 | +0.7 | 21.6 | -1.8 | 32.1 | -0.6 | 65.2 | -0.6 | -0.7 |
| Qwen 3 4B | SFT | 100% | 91.4 | 0.0 | 83.0 | 0.0 | 88.4 | 0.0 | 37.0 | 0.0 | 64.8 | 0.0 | 77.1 | 0.0 | 0.0 |
| | LoRA | 0.41% | 91.0 | -0.4 | 84.6 | +1.6 | 88.8 | +0.4 | 37.6 | +0.6 | 66.7 | +1.9 | 78.0 | +0.9 | +0.8 |
| | ReFT | 0.04% | 90.5 | -0.9 | 64.9 | -18.1 | 84.7 | -3.7 | 37.7 | +0.7 | 66.8 | +2.0 | 68.7 | -8.4 | -4.7 |
| | Ours | 0.04% | 90.7 | -0.7 | 80.6 | -2.4 | 88.6 | +0.2 | 37.4 | +0.4 | 65.0 | +0.2 | 74.0 | -3.1 | -0.9 |
| Llama-3.1-8B | SFT | 100% | 91.6 | 0.0 | 86.4 | 0.0 | 80.6 | 0.0 | 44.0 | 0.0 | 47.7 | 0.0 | 67.2 | 0.0 | 0.0 |
| | LoRA | 0.26% | 92.3 | +0.7 | 88.8 | +2.4 | 80.4 | -0.2 | 43.8 | -0.2 | 45.4 | -2.3 | 67.6 | +0.4 | +0.1 |
| | ReFT | 0.02% | 91.3 | -0.3 | 82.1 | -4.5 | 77.9 | -2.7 | 40.1 | -3.9 | 45.4 | -2.3 | 64.4 | -2.8 | -2.8 |
| | Ours | 0.02% | 92.3 | +0.7 | 87.2 | +0.8 | 80.4 | -0.2 | 43.4 | -0.4 | 47.6 | +0.1 | 69.1 | +1.9 | +0.5 |

Table 2. Comparison with methods with tiny parameter budgets. Δ is performance difference relative to SFT.

| Model | Method | Params (%) | BoolQ | Δ | WinoG | Δ | ARC-C | Δ | GSM8K | Δ | AQuA | Δ | ListOps | Δ | Avg. |
|---|---|---|---|---|---|---|---|---|---|---|---|---|---|---|---|
| Llama-3.2-1B | LoFIT | 0.003% | 83.0 | -5.2 | 49.2 | -9.0 | 46.3 | -20.0 | 28.2 | -4.0 | 32.5 | -3.7 | 64.1 | -2.1 | -7.3 |
| | JoLA | 0.007% | 83.6 | -4.6 | 48.7 | -9.5 | 52.5 | -7.8 | 28.0 | -4.2 | 31.1 | -5.1 | 62.2 | -4.0 | -5.9 |
| | Ours (1-vec) | 0.003% | 86.1 | -2.1 | 57.6 | -0.6 | 58.0 | -1.7 | 28.6 | -3.6 | 30.9 | -6.3 | 64.6 | -1.6 | -2.6 |
| | Ours r=1 | 0.005% | 86.2 | -2.0 | 52.0 | -6.2 | 58.8 | -0.9 | 29.2 | -3.0 | 33.6 | -2.6 | 64.4 | -1.8 | -2.8 |
| gemma-3-1b | LoFIT | 0.003% | 75.8 | -7.5 | 49.6 | -1.8 | 46.7 | -1.5 | 15.1 | -8.3 | 28.4 | -4.3 | 62.3 | -3.5 | -4.5 |
| | JoLA | 0.007% | 75.1 | -8.2 | 51.2 | -0.2 | 46.4 | -1.8 | 14.7 | -8.7 | 26.5 | -6.3 | 57.8 | -8.0 | -5.5 |
| | Ours (1-vec) | 0.003% | 74.2 | -9.1 | 51.4 | 0.0 | 44.5 | -3.7 | 18.0 | -5.4 | 26.8 | -5.9 | 61.9 | -3.9 | -4.7 |
| | Ours r=1 | 0.005% | 83.1 | -0.2 | 51.8 | +0.4 | 49.0 | +0.8 | 19.0 | -4.4 | 27.4 | -5.3 | 65.2 | -0.6 | -1.5 |
| Qwen 3 4B | LoFIT | 0.003% | 90.2 | -1.2 | 77.8 | -5.2 | 87.6 | -0.8 | 39.1 | +2.1 | 67.8 | +3.0 | 65.3 | -11.8 | -2.3 |
| | JoLA | 0.007% | 89.6 | -1.8 | 75.7 | -7.3 | 88.1 | -0.3 | 38.9 | +1.9 | 64.0 | -0.8 | 64.5 | -12.6 | -3.5 |
| | Ours (1-vec) | 0.003% | 90.4 | -1.0 | 79.0 | -4.0 | 86.3 | -2.1 | 37.1 | +0.1 | 63.4 | -1.4 | 65.3 | -11.8 | -3.4 |
| | Ours r=1 | 0.005% | 90.0 | -1.4 | 80.2 | -2.8 | 88.4 | 0.0 | 39.1 | +2.1 | 65.9 | +1.1 | 63.6 | -13.5 | -2.4 |
| Llama-3.1-8B | LoFIT | 0.001% | 89.9 | -1.7 | 48.7 | -39.1 | 73.8 | -6.8 | 39.8 | -4.2 | 47.0 | -0.7 | 64.8 | -2.4 | -9.2 |
| | JoLA | 0.003% | 90.5 | -1.1 | 50.3 | -37.5 | 76.9 | -3.7 | 37.7 | -6.3 | 46.2 | -1.5 | 63.8 | -3.4 | -8.9 |
| | Ours (1-vec) | 0.001% | 91.7 | +0.1 | 86.6 | +0.2 | 80.0 | -0.6 | 39.7 | -4.3 | 44.9 | -2.8 | 66.4 | -2.8 | -1.7 |
| | Ours r=1 | 0.003% | 91.3 | -0.6 | 85.9 | -0.5 | 80.3 | -0.3 | 40.7 | -3.3 | 45.6 | -2.1 | 68.5 | +1.3 | -0.9 |

Additionally, the gradient flow dynamics with the MSE loss can be approximated as

$$\dot{\delta h} \approx RY^{\top} \quad \dot{\delta W} \approx RF^{\top}.$$

So, up to higher order error, $\delta h$ and $\delta W$ will simply be $\alpha RY^{\top}$ and $\alpha RF^{\top}$ for some $\alpha$. Since these left singular vectors are aligned, the dominant directions these parameters learn to match are the same. □

The relationship between the left-singular vectors of $RY^{\top}$ and $RF^{\top}$ is not guaranteed. In fact, since $F$ is an arbitrary function, these two can have incredibly dissimilar dominant singular vector spaces. However, for real data such as Winogrande (Sakaguchi et al., 2021), Figure 3 shows the inner products for the top 50 singular vectors of these matrices at three layers of the Llama model being trained.

Notably, these figures are nearly diagonal, meaning that these top left-singular subspaces are highly aligned. This leads to an important problem: steering and fine-tuning are going to approximate the same update, at least with only small updates. This, of course, is undesirable, since learning different subspaces would lead to more dimensions of the residual matrix $R$ being learned.

## 5. Principled Steering and Joint Adaptation

Building on the theoretical framework established in Section 3 and 4, we now translate our analysis of weight-activation equivalence (and their differences) **into actionable algorithmic innovations**. First, our derivation of the mapping between spaces identifies the most expressive intervention site within the Transformer block, leading to the **post-block steering** method. Second, the functional differences ob-

served between weight and activation updates motivates a **joint adaptation** regime. In the following subsections, we detail the implementation of these techniques.

### 5.1. Post-Block Steering

We parameterize our intervention as a bottleneck adapter:

$$h \rightarrow h + W_2\phi(W_1 h),$$

where $W_1 \in \mathbb{R}^{r \times d}$ and $W_2 \in \mathbb{R}^{d \times r}$ project the hidden state $h$ into a low-dimensional subspace of rank $r$ and project it back to the model's dimension. We evaluate both linear and non-linear variants, where $\phi$ is either the identity function or a non-linear activation (e.g., SiLU). While structurally similar to ReFT (Wu et al., 2024), we introduce three principled improvements derived from Section 3:

- **Theoretical Foundation of the Locus:** While ReFT identifies the residual stream as a high-performance intervention site through empirical selection, our framework provides a formal theoretical foundation for this choice (Section 3.2).
- **Global Scope:** Our analysis establishes that weight and activation updates equivalence requires global modification, while ReFT treats intervention layers as discrete hyperparameters requiring empirical search. Consequently, we apply the intervention at *every* model layer by default.
- **Bottleneck Adapter Parameterization:** ReFT focuses on a specific low-rank linear parameterizations with constraints such as orthonormal rows. We employ a more flexible bottleneck adapter without these constraints and permits non-linear transformations.

### 5.2. Orthogonality-Constrained Joint Adaptation

Given the tendency for weight and activation updates to converge into the same subspace early in training—resulting in functional redundancy (Section 4.1)—we enforce an orthogonality constraint between the output spaces of the steering intervention and the weight update to prevent this collapse.

Specifically, we define $W_2$ as the output projector for the steering adapter and $B$ as the output projector for the weight update (e.g., the LoRA $B$ matrix). We compute $V$, the orthogonal basis for the column space of $B$, and project $W_2$ onto the orthogonal complement:

$$W_2 \mapsto (I - VV^{\top})W_2.$$

This operation ensures that $W_2$ and $B$ operate in strictly orthogonal subspaces, forcing the activation updates to learn features complementary to the weight updates.

## 6. Experiments

We empirically evaluate the following claims:

*Table 3.* Instruction tuning on Llama 3.1 8B with AlpacaEval 2.0 (length-controlled win rate against GPT-4 Turbo).

| Method | Params (%) | LC Win Rate (↑) | SE |
|---|---|---|---|
| Full SFT | 100% | 11.49 | ±0.51 |
| Ours (nonlinear, $r=16$) | 0.05% | **11.34** | ±0.48 |
| Ours (linear, $r=16$) | 0.05% | 11.00 | ±0.53 |
| LoRA ($r=16$) | 0.52% | 9.59 | ±0.40 |
| LoRA ($r=8$) | 0.26% | 10.52 | ±0.48 |
| ReFT ($r=16$, all layers) | 0.05% | 10.96 | ±0.43 |
| ReFT ($r=4$, 4 layers) | 0.004% | 9.50 | ±0.49 |

*Table 4.* RL on DeepSeek-R1-Distill-Qwen-1.5B with GSM8K.

| Method | Params (%) | Pass@1 |
|---|---|---|
| Base model | - | 10.2 |
| LoRA | 0.52% | $81.5 \pm 0.7$ |
| Ours (nonlinear) | 0.04% | $84.7 \pm 0.5$ |
| Ours (linear) | 0.04% | $84.3 \pm 0.8$ |

- **Principled steering approximates full-parameter fine-tuning (Section 6.1).** Post-block steering performs within a 0.2%–0.9% margin of full-parameter fine-tuning (SFT) while training only 0.04% model parameters.
- **Generalization to complex training paradigms (Section 6.2).** The effectiveness of our approach extends to more complex optimization landscapes like instruction tuning and reinforcement learning (RL).
- **Non-linear parameterization provides marginal utility (Section 6.3).** While adding non-linearity to the activation adapters offers stable gains, the improvement is negligible across ranks and model scales.
- **Jointly adapting both weights and activations is more expressive (Section 6.4).** Joint training in both parameter and activation spaces frequently outperforms individual adaptation methods, accessing a more functional capacity.
- **Orthogonality is crucial for joint training (Section 6.5).** We find that enforcing the orthogonality constraint (described in Section 5.2) is necessary to prevent functional redundancy and access the benefits of joint adaptation.

We provide experiment details in Appendix F.

### 6.1. Approximating full-parameter SFT

**Setup.** We evaluate our approach in two regimes:

1. **Performance-oriented PEFT:** We compare against **LoRA** (Hu et al., 2022), the industry standard for weight-space adaptation, and **ReFT** (Wu et al., 2024), a high-performance representation steering method (Table 1).
2. **Ultra-Efficient Steering:** We benchmark against **LoFiT** (Yin et al., 2024) and **JoLA** (Lai et al., 2025), which target extreme parameter efficiency (Table 2).

The parameter budget (Params (%)) is the percentage of trainable parameters relative to the total model size. To ensure a fair comparison, we adjust our adapter rank to match

the baselines' parameter counts. We allocate LoRA a larger budget (0.45% vs ours 0.04%) because its performance degrades significantly at lower ranks. *Ours (1-vec)* variant in Table 2 denotes a version of our method that trains single steering vectors. All steering adapters in this subsection are linear ($\phi$ is the identity function). For a fair comparison, ReFT is also applied at all layers (matching our global intervention policy).

**Results.** Table 1 shows that our approach maintains an average performance gap within 0.2%–0.9% of SFT while training only 0.04% parameters. On 1B-scale models, we match or outperform LoRA despite training $11\times$ fewer parameters. **Compared to ReFT at the same** 0.04% **budget, our method is significantly more stable on complex tasks**. The performance gap is particularly pronounced on ListOps, a long-range dependency task where attention plays a critical role. While ReFT drops by up to 16.9%, our approach limits the gap to 3.1%.

Table 2 shows that **our approach achieves state-of-the-art among methods with tiny parameter budgets**. With only 0.001%–0.005% trainable parameters, we yield the best average performance across all models. While baselines remain competitive on mid-sized models like Qwen-4B, they exhibit significant instability at larger scales, collapsing to an average gap of approximately $-9\%$ on Llama-3.1-8B. Our approach remains robust, outperforming the best-performing baseline at the 8B scale by a margin of 8.0%.

### 6.2. Generalization to more complex objectives

**Setup.** Next, we evaluate post-block steering on instruction tuning and RL, testing whether our results generalizes from the structured reasoning tasks studied previously to open-ended generation and iterative policy optimization.

For instruction tuning, we fine-tune Llama 3.1 8B on Alpaca-Cleaned (Taori et al., 2023) ($\sim$52K pairs) for 3 epochs. We compare post-block steering against full-parameter SFT, LoRA ($r = 8$ and $r = 16$), and two ReFT variants: the published sparse configuration ($r = 4$ at 4 layers) and a parameter-matched variant ($r = 16$ at all layers). While we report the best of two learning rates for SFT and LoRA, our method uses a fixed learning rate of $2 \times 10^{-3}$. Models are evaluated on AlpacaEval 2.0 (Li et al., 2023b) reporting length-controlled (LC) win rates against GPT-4 Turbo and standard errors (SE) across 805 instructions.

For RL, we perform Group Relative Policy Optimization (GRPO) (Shao et al., 2024) on DeepSeek-R1-Distill-Qwen-1.5B (DeepSeek-AI, 2025), comparing our post-block adapter against the standard LoRA implementation in TRL (von Werra et al., 2020; Schulman & Lab, 2025). Both methods use rank $r = 8$.

*Table 5.* Joint training vs. its components. "Joint" is naive joint training, "J-Orth" adds the orthogonality constraint. Joint uses LoRA (0.45%) + Adapter (0.34%), total 0.79% trainable params. LoRA $r{=}16$ has 0.9% params.

| Model | Data | SFT | LoRA$_{16}$ | LoRA$_8$ | Adpt. | Joint | J-Orth |
|---|---|---|---|---|---|---|---|
| Llama-1B | GSM8K | **32.2** | 31.2 | 31.8 | 31.3 | 31.1 | 31.1 |
| | BoolQ | 88.2 | 88.4 | 84.7 | 86.2 | 88.4 | **88.5** |
| | WinoG | 58.2 | **64.3** | 57.0 | 52.4 | 56.9 | 59.3 |
| Gemma-1B | GSM8K | 23.4 | 23.8 | 22.6 | 21.6 | 22.8 | **24.4** |
| | BoolQ | 83.3 | **84.8** | 84.7 | 81.4 | 84.5 | **84.8** |
| | WinoG | 51.4 | **51.8** | 51.4 | 50.8 | 51.3 | 50.9 |
| Qwen-4B | GSM8K | 37.0 | 38.3 | 37.6 | 37.3 | 37.9 | **38.8** |
| | BoolQ | **91.4** | 91.0 | 91.0 | 90.3 | 90.7 | **91.4** |
| | WinoG | 83.0 | 83.0 | **84.6** | 82.5 | 82.7 | 83.5 |

**Results.** Table 3 summarizes the instruction tuning results. Our nonlinear post-block adapter achieves an LC win rate of 11.34%, nearly closing the gap to full-parameter SFT (11.49%) within 0.15%. Both our linear (11.00%) and nonlinear variants outperform LoRA (10.52%), despite LoRA using more trainable parameters. This confirms that the expressivity of post-block steering extends to open-ended instruction following.

The RL evaluation (Table 4) reveals that our post-block steering outperforms LoRA by up to 3.2% while using $13\times$ fewer parameters (0.04% vs. 0.52%). These results suggest that activation-space interventions utility generalizes from supervised signals to the non-stationary gradients and shifting policy distributions typical of RL optimization.

We also note that the marginal gap between our linear and nonlinear variants on these complex scenarios suggests that while nonlinear parameterization offers peak performance, the linear approximation is highly robust, a trend we examine more broadly next in Section 6.3.

### 6.3. Effect of nonlinearity

**Setup.** A natural concern is whether our first-order analysis holds under practical training conditions. To test this, we examine how non-linearity interacts with increasing rank and model scale using Qwen3 (Yang et al., 2025) model suite (0.6B, 1.7B, 4B, and 8B) on Winogrande.

**Results.** Figure 4 illustrates the interaction between non-linearity, rank, and model scale. Across models 1.7B and larger, nonlinear adapters perform comparably to their linear counterparts, and performance remains stable across ranks. This suggests that **linear shifts are largely sufficient, supporting the validity of our first-order analysis**, and that intervention site matters more than adapter capacity. At 0.6B, we observe performance degradation and increased variance at higher ranks, indicating that smaller models may be more sensitive to overparameterization.

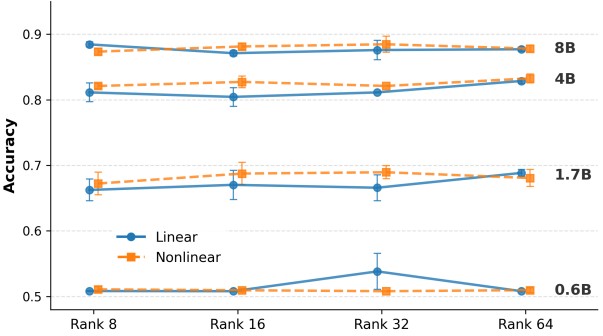

*Figure 4.* Effect of rank and linearity across model sizes

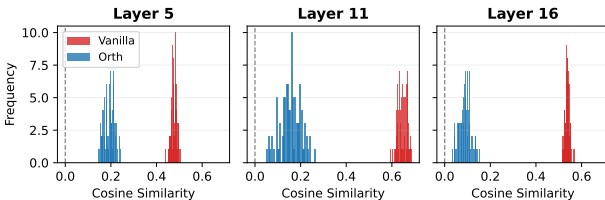

*Figure 5.* Distribution of cosine similarity between weight and adapter shifts. Closer to zero indicates more orthogonal solutions.

## 6.4. Joint Training is more expressive

**Setup.** We compare our joint training with orthogonality constraint (**Joint-Orth**) against several baselines: full-parameter **SFT**, individual **LoRA** ($r = 8$ and $r = 16$), an individual **Activation Adapter** ($r = 64$), and naive joint training without the constraint (**Joint**). To ensure a balanced contribution from both weight and activation spaces, we set the adapter rank such that its parameter count is comparable to LoRA's (Activation Adapter: $0.34\%$ vs. LoRA: $0.45\%$).

**Results.** Table 5 shows that Joint-Orth often matches or exceeds the strongest individual baselines and full-parameter SFT. Joint-Orth frequently outperforms LoRA $r = 16$ despite having a smaller total parameter footprint ($0.79\%$ vs $0.90\%$), particularly on reasoning-heavy tasks like GSM8K. The value of the orthogonality constraint is apparent when compared to naive Joint training, which frequently underperforms its strongest individual component, providing empirical support for the functional redundancy predicted in Proposition 4.2. When Joint-Orth does not improve performance (e.g., WinoG on Qwen), it remains competitive with the top-performing baseline.

In reasoning-heavy tasks (e.g., GSM8K, BoolQ), the performance boost from **Joint-Orth** suggests that weight updates and activation steering target complementary functional roles (e.g., knowledge retrieval and procedural logic). By enforcing orthogonality, we enable these spaces to decouple. Conversely, for linguistic tasks like WinoGrande, weight and activation interventions likely target redundant functional paths, limiting additive gains.

## 6.5. Orthogonality unlocks joint training benefits

**Setup.** To test the effect of the orthogonality constraint, we plot the distribution of cosine similarities between the steering shift and the weight update shift. We randomly sample 100 prompts from the Winogrande dataset and evaluate on Llama-3.2-1B. For each layer, we compute (1) the activation adapter's output shift and (2) the LoRA-induced shift, measured as the difference in MLP output with and without the LoRA update. We then compute the cosine similarity between these two shifts.

**Results.** Figure 5 shows that the naively trained joint model (vanilla, red) shows high correlation between the activation adapter shift and the LoRA shift, confirming the learning-the-same-subspace hypothesis from Section 4.1. In contrast, the orthogonally projected case (blue) shows very small cosine similarity. While it is tempting to interpret this lack of correlation as the projection nullifying the adapter's contribution, we find the 2-norm of the $W_2$ matrices range from $0.8$ to $1.4$ across layers, demonstrating that the adapter retains significant magnitude (Figure 6 in Appendix G.1).

## 7. Conclusion

We transitioned activation steering from a heuristic-driven black-box to a principled framework by establishing a first-order equivalence between weight-space updates and activation-space interventions. We identify post-block output as a theoretically grounded, highly expressive intervention point, achieving steering accuracy within $0.2\%$–$0.9\%$ of full-parameter fine-tuning while training only $0.04\%$ of parameters. We show that weight and activation updates are functionally complementary, and introduce a joint adaptation regime that surpasses the performance ceilings of either method in isolation, unlocking a new pathway for adapting large-scale models in memory-constrained environments.

**Limitations.** Our theoretical link between weight updates and activation interventions is local and first-order, and may break when perturbations are large or higher-order effects dominate. Empirically, our gains combine a post-block locus with a global intervention policy (across layers), so current results do not fully disentangle where to intervene from how to intervene. Our evaluation is limited to a moderate set of tasks/models and primarily reports accuracy/parameter count; we do not comprehensively study latency/memory overhead or broader side effects (e.g., distribution shift).

## Impact Statement

This paper presents work whose goal is to advance the field of Machine Learning. There are many potential societal consequences of our work, none which we feel must be

specifically highlighted here.

## Acknowledgments

We are grateful for the support of the National Science Foundation (NSF) (CCF2106707), the Defense Advanced Research Projects Agency (DARPA Young Faculty Award), and the Wisconsin Alumni Research Foundation (WARF).

We thank Sprocket Lab members for the meaningful feedback on this manuscript.

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

## A. Notation and Shapes

| Symbol | Meaning |
|--------|---------|
| $\delta A$ | A learned parameter with name $A$ |
| $\Delta A$ | A small change in $A$ |
| $h$ | The input to the MLP block |
| $d_{\text{model}}$ | The embedding dimension of the transformer |
| $d_{\text{mlp}}$ | The hidden dimension of each MLP layer |
| $X$ | Data matrix |
| $Y$ | Base MLP output |
| $G$ | Target module output |
| $\phi(\cdot)$ | Some non-linearity |
| $\mathcal{C}(\cdot)$ | The column space operator |
| $\mathcal{R}(\cdot)$ | The row space operator |
| pre-MLP | Updates to the input of the MLP |
| post-MLP | Updates to the output of the MLP |
| post-block | Updates to the residual stream, after the skip-connection |

*Table 6.* Notation

Let $h \in \mathbb{R}^{d_{\text{model}}}$ denote the input to the MLP block (e.g., the post-attention hidden at a given token). We write

$$a = W_1 h \in \mathbb{R}^{d_{\text{mlp}}}, \qquad m = \phi(a) \in \mathbb{R}^{d_{\text{mlp}}}, \qquad y = W_2 m \in \mathbb{R}^{d_{\text{model}}},$$

with weight matrices

$$W_1 \in \mathbb{R}^{d_{\text{mlp}} \times d_{\text{model}}}, \qquad W_2 \in \mathbb{R}^{d_{\text{model}} \times d_{\text{mlp}}}.$$

The nonlinearity $\phi$ acts elementwise; we denote its elementwise derivative by $\phi'(a)$ and write $\text{Diag}(\phi'(a))$ for the diagonal matrix whose diagonal is $\phi'(a)$.

During fine-tuning we obtain updated weights $W_1 + \delta W_1$ and $W_2 + \delta W_2$. We define the *fine-tuning effect* at $h$ as

$$t(h) \stackrel{\text{def}}{=} y_{\text{FT}}(h) - y(h), \qquad y_{\text{FT}}(h) = (W_2 + \Delta W_2)\, \phi\big((W_1 + \Delta W_1)h\big).$$

We derive $t(h)$ to first order in $(\Delta W_1, \Delta W_2)$.

## B. Gradient Derivations

### B.1. Simple MLP (no gating)

Starting from $y(h) = W_2\, \phi(W_1 h)$, define

$$a = W_1 h, \qquad \Delta a = W_1 \Delta h.$$

A first-order Taylor expansion of $\phi$ around $a$ yields

$$\phi(a + \Delta a) = \phi(a) + \text{Diag}\big(\phi'(a)\big)\, \Delta a \; + \; \mathcal{O}\big(\|\Delta a\|^2\big).$$

Thus

$$y(h + \Delta h) = W_2\, \phi\big(W_1(h + \Delta h)\big) = W_2\, \phi(a + \Delta a)$$
$$= W_2\Big[\phi(a) + \text{Diag}\big(\phi'(a)\big)\, \Delta a\Big] \; + \; \mathcal{O}\big(\|\Delta a\|^2\big).$$

Subtracting $y(h) = W_2\phi(a)$ and substituting $\Delta a = W_1 \delta h$, we obtain

$$\boxed{\Delta y_{\text{steer}}(h) \; \approx \; W_2\, \text{Diag}\big(\phi'(a)\big) W_1\, \Delta h \; = \; A(h)\, \Delta h,}$$

where the *pre-MLP Jacobian* is

$$\boxed{A(h) \; = \; W_2\, \text{Diag}\big(\phi'(W_1 h)\big)\, W_1 \; \in \; \mathbb{R}^{d_{\text{model}} \times d_{\text{model}}}.}$$

All neglected terms are $\mathcal{O}(\|\delta h\|^2)$ (second order).

**GLU (gated) MLP**

For GLU blocks (as in LLaMA/Gemma/Qwen),

$$a_g = W_g h, \qquad a_u = W_u h, \qquad m = \phi(a_g) \odot a_u, \qquad y = W_d m,$$

with

$$W_g, W_u \in \mathbb{R}^{d_{\text{mlp}} \times d_{\text{model}}}, \qquad W_d \in \mathbb{R}^{d_{\text{model}} \times d_{\text{mlp}}}.$$

Under $h \mapsto h + \delta h$,

$$\Delta a_g = W_g \Delta h, \qquad \Delta a_u = W_u \Delta h.$$

A first-order Taylor expansion of $m = \phi(a_g) \odot a_u$ gives

$$\Delta m \;=\; \big(\phi'(a_g) \odot a_u\big) \odot \Delta a_g \;+\; \phi(a_g) \odot \Delta a_u \;+\; \mathcal{O}\big(\|\Delta h\|^2\big).$$

Therefore

$$\Delta y_{\text{steer}}(h) = W_d \, \Delta m$$
$$\approx W_d \Big[ \underbrace{\big(\phi'(a_g) \odot a_u\big) \odot (W_g \Delta h)}_{\text{gate path}} \;+\; \underbrace{\phi(a_g) \odot (W_u \Delta h)}_{\text{value path}} \Big].$$

Equivalently, factor as a linear map in $\Delta h$:

$$\boxed{\Delta y_{\text{steer}}(h) \;\approx\; A_{\text{GLU}}(h) \, \Delta h, \quad A_{\text{GLU}}(h) = W_d \Big[ \text{Diag}\big(\phi(a_g)\big) W_u \;+\; \text{Diag}\big(a_u \odot \phi'(a_g)\big) W_g \Big].}$$

All neglected terms are $\mathcal{O}(\|\Delta h\|^2)$.

**LayerNorm (optional)**

If a (pre-)LayerNorm precedes the MLP, write $h = \text{LayerNorm}(\tilde{h})$ with Jacobian $J_{\text{LayerNorm}}(\tilde{h})$. A pre-MLP steering perturbation in $\tilde{h}$ yields

$$\Delta y_{\text{steer}}(\tilde{h}) \;\approx\; A\big(\text{LayerNorm}(\tilde{h})\big) J_{\text{LayerNorm}}(\tilde{h}) \, \Delta \tilde{h} \quad \text{(simple MLP)},$$

or

$$\Delta y_{\text{steer}}(\tilde{h}) \;\approx\; A_{\text{GLU}}\big(\text{LayerNorm}(\tilde{h})\big) J_{\text{LayerNorm}}(\tilde{h}) \, \Delta \tilde{h} \quad \text{(GLU)}.$$

## Second-order remainder

All formulas above are first-order in $\Delta h$. The neglected terms scale as $\mathcal{O}(\|\Delta h\|^2)$, so keeping $\|\Delta h\|$ small preserves the validity of the linear approximation. In practice, this aligns with using small intervention scales and/or regularizers that keep edits in the first-order regime.

**Elementwise (Hadamard) view.** Equivalently,

$$\Delta y_{\text{steer}}(h) \;\approx\; W_d \big[ \big(\phi'(a_g) \odot a_u\big) \odot (W_g \, \delta h) \;+\; \phi(a_g) \odot (W_u \, \delta h) \big].$$

**Shapes.** $D_g, D_u \in \mathbb{R}^{d_{\text{mlp}} \times d_{\text{mlp}}}$ are diagonal row-scalers; $J_m(h) \in \mathbb{R}^{d_{\text{mlp}} \times d_{\text{model}}}$; $A(h) \in \mathbb{R}^{d_{\text{model}} \times d_{\text{model}}}$; and $\Delta y_{\text{steer}}(h) \in \mathbb{R}^{d_{\text{model}}}$.

## C. Expressivity

### C.1. The Specific Location

Other methods steer the outputs of the MLP or the attention modules before the skip-connections are added. In notation, for a steering vector $\delta h$,

$$y(h) = h + \text{Attn}(h) + (\text{GLU}(h + \text{Attn}(h)) + \delta h)$$

in contrast to the (minorly different) form

$$y(h) = h + \text{Attn}(h) + \text{GLU}(h + \text{Attn}(h)) + \delta h$$

In principle, these can be identical for a completely freely parameterized $\delta h$. However, there is an important difference which is that $\delta h$ can only depend on parts of the model and not others. Post MLP methods location has $\delta h(\text{GLU}(h + \text{Attn}(h)))$ while our method has $\delta h(y(h))$. To investigate this further in a clean way, take both steering vectors as a linear update from the existing hidden state, i.e.

$$\delta h_p = A_p h$$

$$\delta h_{\text{steer}} = A_{\text{steer}} h$$

**Proposition C.1.** *Let* $\mathcal{D} = \{h_i\}_{i \in \mathcal{I}}$ *be a collection of hidden states, and let the following be linear subspaces of functions of* $\mathcal{D}$*:*

$$A = \text{span}\{h + \text{Attn}(h) | h \in \mathcal{D}\}$$

$$B = \text{span}\{\text{GLU}(h + \text{Attn}(h)) | h \in \mathcal{D}\}$$

*Note that these do not necessarily have to coincide, based on the the down projection* $W_d$ *in the* GLU*. In fact, assume that* $A \cap B = \{0\}$*, and denote the projections onto these subspaces, remove all components from the other as* $P_A$ *and* $P_B$*. Then, for any* $A_p$*,*

$$A_{\text{steer}} = A_p P_B$$

*will satisfy that*

$$y_p(h) = y_{\text{steer}}(h)$$

*for every* $h \in \mathcal{D}$*.*

*Proof.* This is a straightforward application of projections. For $v \in B$, $P_B v = v$, and for $v \in A$, $P_B v = 0$. Thus,

$$
\begin{aligned}
y(h) &= h + \text{Attn}(h) + (\text{GLU}(h + \text{Attn}(h)) + \delta h_{\text{steer}} \\
&= h + \text{Attn}(h) + (\text{GLU}(h + \text{Attn}(h)) + A_{\text{steer}}(h + \text{Attn}(h) + (\text{GLU}(h + \text{Attn}(h)))) \\
&= h + \text{Attn}(h) + (\text{GLU}(h + \text{Attn}(h)) + A_p P_B(h + \text{Attn}(h)) + A_p P_B(\text{GLU}(h + \text{Attn}(h))) \\
&= h + \text{Attn}(h) + (\text{GLU}(h + \text{Attn}(h)) + A_p(\text{GLU}(h + \text{Attn}(h))) \\
&= h + \text{Attn}(h) + (\text{GLU}(h + \text{Attn}(h) + \delta h_p)
\end{aligned}
$$

$\square$

In contrast, notice that for $W_d = 0$ in the GLU, then $\text{GLU}(h + \text{Attn}(h)) = 0$, thus for any input-dependent $\delta h$ only depending on the output of the GLU can no longer depend on the input. Instead, it is forced to be a constant, fixed vector. In any situation like this, steering after the skip-connection is strictly more expressive than steering before it.

The above proposition is quite strong though, and does not hold for the much-more-common full rank $A$ and/or $B$. It is possible to make a statement about the relative error

**Theorem C.2.** *Let* $V$ *and* $V'$ *be the right-singular matrices of* $h + \text{Attn}(h) + \text{GLU}(h + \text{Attn}(h))$ *and* $A_p \text{GLU}(h + \text{Attn}(h))$ *respectively, and let* $Y^{pa}, Y^{mlp}$ *be defined as* $Y_i^{mlp} := h_i + \text{Attn}(h_i)$ *and* $Y_i^{pa} := \text{GLU}(Y_i^{mlp})$*. Then, the optimal relative error satisfies*

$$\min_A \frac{\|A(Y^{pa} + Y^{mlp}) - A_p Y^{pa}\|_F^2}{\|A_p Y^{pa}\|_F^2} = \sum_{i=1}^{d} \left( \frac{(\sigma')_i^2}{\sum_{j=1}^{d}(\sigma')_j^2} \right) \sin^2 \theta_i$$

*where* $\sigma_i$ *is the* $i$*-th singular value of* $A_p Y^{pa}$ *and* $\theta_i$ *is the* $i$*-th principle angle between* $V$ *and* $V'$*.*

*Proof.* For some $A_p$, the best possible steering matrix $A_{\text{steer}}$ will be

$$A_{\text{steer}} = \underset{A}{\text{argmin}} \, \mathcal{L}(A) = \underset{A}{\text{argmin}} \, \frac{\|A(Y^{pa} + Y^{mlp}) - A_p Y^{pa}\|_F^2}{\|A_p Y^{pa}\|_F^2}$$

Taking a gradient and setting it to zero,

$$0 = 2A(Y^{pa} + Y^{mlp})(Y^{pa} + Y^{mlp})^\top - 2A_p Y^{pa}(Y^{pa} + Y^{mlp})^\top$$
$$A = A_p Y^{pa}(Y^{pa} + Y^{mlp})^\top((Y^{pa} + Y^{mlp})(Y^{pa} + Y^{mlp})^\top)^{-1}$$

Placing this in the loss,

$$\mathcal{L}(A_{\text{steer}}) = \frac{1}{n}\|A_p Y^{pa}((Y^{pa} + Y^{mlp})^\top((Y^{pa} + Y^{mlp})(Y^{pa} + Y^{mlp})^\top)^{-1}(Y^{pa} + Y^{mlp}) - I)\|_F^2$$

If we write the reduced SVD of $Y^{pa} + Y^{mlp}$ as

$$U\Sigma V^\top = Y^{pa} + Y^{mlp},$$

then the above reduces significantly to

$$\mathcal{L}(A_{\text{steer}}) = \frac{\|A_p Y^{pa}(I - VV^\top)\|_F^2}{\|A_p Y^{pa}\|_F^2}$$

That is, the optimal error is related to how close the row space of $Y^{pa}$ is in the row space of $Y^{pa} + Y^{mlp}$ and thus is projected away. If we let

$$U'\Sigma'(V')^\top = A_p Y^{pa}$$

be the SVD of $A_p Y^{pa}$, we can decompose the numerator of $\mathcal{L}$ as

$$\|A_p Y^{pa}(I - VV^\top)\|_F^2 = \sum_{i=1}^d (\sigma')_i^2 \|(I - VV^\top)V_i'\|_2^2$$
$$= \sum_{i=1}^d (\sigma')_i^2 \sin^2 \theta_i$$

where $\theta_i$ is the $i$-th principle angle between the subspaces spanned by $V$ and $V'$. If the top $r$ singular vectors of $V'$ span the same space as $V$, then $\sin \theta_i = 0$ for $i = 1, \ldots, r$. Therefore,

$$\mathcal{L}(A_{\text{steer}}) = \sum_{i=1}^d \left(\frac{(\sigma')_i^2}{\sum_{j=1}^d (\sigma')_j^2}\right) \sin^2 \theta_i$$

$\square$

## C.2. Joint vs. Individual Training

Now that it is established that steering on the residual stream is more expressive than inside the MLP, we now turn to understanding how training both with steering and fine-tuning work together.

**Proposition C.3.** *Fix the feature map $F$ and let*

$$\hat{g}(h) = (I + \delta h)(h + (W_2 + \delta W)F(h))$$

*be the hybrid weight-update and post-block steering of $g_{\text{base}}$. Then, $\hat{g}_i$ can represent almost any linear combination of the inputs $\{h_j\}$ and the features $\{F_j\}$. Furthermore, if either $\delta h = 0$ or $\delta W = 0$, which correspond to fine-tuning and steering, respectively, then arbitrary linear combinations become impossible.*

*Proof.* Begin with the negative results. If $\delta h = 0$, then $\hat{g}_i$ must have the form

$$\hat{g}_i(h) = h_i + \sum_j \alpha_j F_j(h)$$

So, unless $F_j$ is a linear map, then $\hat{g}_i$ can only contain a term from the input directly of the form $h_i$, hence any scaling of $h_i$ or any other influence from $h_j$ is missing.

On the other hand, if $\delta W = 0$, then $f'_i$ must have the form

$$\hat{g}_i(h) = \sum_j \alpha_j(h_j + F_j(h))$$

In this situation, the sum of the input with the features is coupled. Therefore, $\hat{g}_i$ cannot contain arbitrary linear combinations, but rather only linear combinations where the feature and skip-connection indices agree.

For the positive case, let $\hat{G}$ have its original form with both a weight update and steering, and assume that we want

$$\hat{g}_i(h) = \sum_j \alpha_{ij} h_j + \beta_{ij} F_j(h)$$

Let $A$ and $B$ be the matrices containing these coefficients. To satisfy the $\alpha$ component of the sum, set $\delta h = A - I$.

Almost always, $A - I$ will be invertible. To satisfy the $\beta$ component of the sum, set $\delta W = (A - I)^{-1}B - W_2$. $\qquad\qquad\square$

## D. Orthogonality

The hope here is to show that without the orthogonality constraint, in a two-layer model, these two different updates will learn in the same subspace.

Let $X$ be the data matrix, $G$ be the matrix of outputs, and

$$\hat{G}(X) = (I + \delta h)(X + (W + \delta W)F(X))$$

where $F(X)$ is some feature map. The dependence on $X$ will often be dropped for notation simplicity. Also, since it will be frequently used, let

$$Y = X + WF$$

and

$$R = G - \hat{G}$$

The question becomes what the gradients w.r.t. $\delta h$ and $\delta W$ are, and how they evolve through gradient descent/flow. Let the loss be a standard MSE loss, i.e.

$$\mathcal{L}(\delta h, \delta W) = \frac{1}{2}\|R\|_F^2$$

where $G$ is some target we are trying to match. The gradients of this are

$$\frac{\partial \mathcal{L}}{\partial(\delta h)} = -R(Y + (\delta W)F)^\top$$

$$\frac{\partial \mathcal{L}}{\partial(\delta W)} = -(I + \delta h)^\top R F^\top$$

The gradient flow for this system will be simplified as a row vector of the parameters:

$$[\dot{\delta h}, \dot{\delta W}] = [RY^\top + RF^\top(\delta W)^\top, RF^\top + (\delta h)^\top RF^\top]$$

Let the initial states of these matrices be $\delta h_0 = 0$ and $\delta W_0 = 0$. For the following, we will need the singular value decomposition of $Y$:

$$U\Sigma V^\top = Y$$

**Proposition D.1.** *Under gradient descent, both $\delta h$ and $\delta W$ will always have their column spaces within the column space of $U$. Additionally, the row-space of $\delta h$ is constrained to be within the column space of $U$.*

*Proof.* For $t = 0$, both matrices are 0 hence we are done.

For $t > 0$, assume this holds for $t - 1$. Then

$$
\begin{aligned}
\mathcal{C}(\delta h_t) &= \mathcal{C}(\delta h_{t-1} - \eta \frac{\partial \mathcal{L}}{\partial (\delta h)}) \\
&\subseteq \mathcal{C}(\delta h_{t-1} - \eta A - \eta \delta h_{t-1} - \eta \delta W_{t-1}) \\
&\subseteq \mathcal{C}(U)
\end{aligned}
$$

Many of the cross terms cancel when looking at the right singular vectors. Additionally,

$$
\begin{aligned}
\mathcal{R}(\delta h_t) &= \mathcal{C}(\delta h_t^\top) \\
&= \mathcal{C}(\delta h_{t-1}^\top - \eta \frac{\partial \mathcal{L}}{\partial (\delta h)}^\top) \\
&\subseteq \mathcal{C}(\delta h_{t-1}^\top - \eta A - \eta \delta W_{t-1}) \\
&\subseteq \mathcal{C}(U)
\end{aligned}
$$

Similarly,

$$
\begin{aligned}
\mathcal{C}(\delta W_t) &= \mathcal{C}(\delta h_{t-1} - \eta \frac{\partial \mathcal{L}}{\partial (\delta W)}) \\
&\subseteq \mathcal{R}(-\eta \delta h_{t-1}) \cup \mathcal{C}(\delta W_{t-1} - \eta A - \eta \delta W_{t-1}) \\
&\subseteq \mathcal{C}(U)
\end{aligned}
$$

$\square$

The above proposition indicates that the parameters $\delta W$ and $\delta h$ learn in the correct subspace if $Y$ is low-rank. This, however, does not tell us how the directions within these spaces are learned, and if when the updates are constrained to be low-rank, which directions will be learned.

**Theorem D.2.** *When learning the function*

$$
\hat{g}(x) = (I + \delta h)(x + (W + \delta W)F(x))
$$

*with gradient descent and MSE loss, early in training, $\delta h$ and $\delta W$ will align with the dominant singular values of $RY^\top$ and $RF^\top$ respectively.*

*Proof.* Let $X$ be the matrix of inputs, $G$ the matrix of labels, $\hat{G}$ the matrix of outputs, $Y = X + WF$, and $F$ the matrix of features $F(x)$ for each input $x$ in $X$, respectively. Early in training, when both $\delta h$ and $\delta W$ are close to zero, the product $\delta h \delta W$ can be neglected. As such, we can approximate

$$
\hat{g}(x) \approx x + (W + \delta W)F(x) + \delta h(x + WF(x))
$$

In matrix form,

$$
\hat{G} \approx X + (\delta W)F + (\delta h)Y
$$

From this, the early gradient flow dynamics can be computed:

$$\dot{\delta h} = (\hat{G} - G)Y^\top = (\hat{G} - X)Y^\top - ((\delta W)F + (\delta h)Y)Y^\top$$
$$\dot{\delta W} = (\hat{G} - G)F^\top = (\hat{G} - X)F^\top - ((\delta W)F + (\delta h)Y)F^\top$$

or equivalently,

$$\begin{bmatrix} \dot{\delta h} & \dot{\delta W} \end{bmatrix} = \begin{bmatrix} (\hat{G} - G)Y^\top & (\hat{G} - G)F^\top \end{bmatrix} - \begin{bmatrix} \delta h & \delta W \end{bmatrix} \begin{bmatrix} YY^\top & YF^\top \\ FY^\top & FF^\top \end{bmatrix}$$

To simplify notation, let

$$A = \begin{bmatrix} YY^\top & YF^\top \\ FY^\top & FF^\top \end{bmatrix}$$
$$B = \begin{bmatrix} (\hat{G} - G)Y^\top & (\hat{G} - G)F^\top \end{bmatrix}$$

This system has a simple analytical solution. Let $U\Lambda U^\top$ be the eigendecomposition of $U$ with an orthonormal basis for $U$ (which is guarantees from $A$ being symmetric).

$$\begin{bmatrix} \delta h & \delta W \end{bmatrix}(t) = -BU\Lambda^{-1}(e^{t\Lambda} - 1)U^\top$$

Aside from this, noting the gradient flow of both parameters, when $t \approx 0$, it can be seen that the learning is simply

$$\dot{\delta h} \approx (\hat{G} - G)Y^\top$$
$$\dot{\delta W} \approx (\hat{G} - G)F^\top$$

So for very early in training, the learned left singular directions are going to be the dominant singular directions of $(\hat{G} - G)Y^\top$ and $(\hat{G} - G)F^\top$. Therefore, if these directions mostly agree with each other, the two matrices will learn to match $G$ in the same subspaces. $\qquad\square$

## E. Error Propagation

This section is meant to be useful for analyzing the error when a model with target parameters is known and is attempted to be mimicked. This contains a loose collection of facts about this error control to be used as-needed rather than for a core result in this work.

Regardless of the steering method chosen, one method for this to succeed is to mimic the oracle (i.e. the fine-tuned hidden states) with sufficient accuracy that it damps errors rather than amplifying them. To this end, models with different but similar parameters are considered, which will lead to a condition which guarantees close performance between the two models.

**Lemma E.1.** *Let $h, h' \in \mathbb{R}^n$ be such that $\|h - h'\| \leq \epsilon$. Then, the effects of a layer norm are as follows:*

$$\|\mathrm{LayerNorm}(h) - \mathrm{LayerNorm}(h')\|_2 \leq \frac{\epsilon}{\min\{\mathrm{std}(h), \mathrm{std}(h')\}}$$

*Proof.* Note that LayerNorm can be written as $\mathrm{LayerNorm}(h) = \frac{\sqrt{d}Ph}{\|Ph\|_2}$ where $P = I - \frac{1}{d}\mathbf{1}\mathbf{1}^\top$. Note that $P$ is a projection,

so $\|P\|_2 = 1$. Thus,

$$\begin{aligned}
\|\text{LayerNorm}(h) - \text{LayerNorm}(h')\|_2 &= \sqrt{d} \left\| \frac{Ph}{\|Ph\|_2} - \frac{Ph'}{\|Ph'\|_2} \right\|_2 \\
&\leq \sqrt{d} \frac{\|Ph - Ph'\|_2}{\min\{\|Ph\|_2, \|Ph'\|_2\}} \\
&= \frac{\|Ph - Ph'\|_2}{\min\{\text{std}(h), \text{std}(h')\}} \\
&\leq \frac{\|P\|_2 \|h - h'\|_2}{\min\{\text{std}(h), \text{std}(h')\}} \\
&= \frac{\epsilon}{\min\{\text{std}(h), \text{std}(h')\}}
\end{aligned}$$

$\square$

This is a well established lemma, and allows for the behavior of errors between two hidden vectors to be bounded when passed through an MLP.

**Lemma E.2.** *Let* $\|h - h'\|_2 \leq \epsilon$, $\|h\|_2 = \|h'\|_2 = \sqrt{n}$, *and* $\|W - W'\|_2 \leq \delta$. *Then*

$$\|Wh - W'h'\|_2 \leq \|W\|_2 \epsilon + \sqrt{n}\delta$$

*Proof.*

$$\begin{aligned}
\|Wh - W'h'\|_2 &= \|Wh - Wh' + Wh' - W'h'\|_2 \\
&\leq \|W\|_2 \|h - h'\|_2 + \|W - W'\|_2 \|h'\|_2 \\
&\leq \|W\|_2 \epsilon + \sqrt{n}\delta
\end{aligned}$$

$\square$

**Lemma E.3.** *Consider the following function:*

$$y(h) = \text{GLU}(h) := h + W_d(\sigma(W_g \text{LayerNorm}(h)) \odot W_u \text{LayerNorm}(h))$$

*This represents the output of a GLU with layer norm and skip connection. Denote* $y'(h)$ *to be a similar GLU with different parameters. Assume that* $\|W_* - W'_*\|_2 \leq \delta$ *for each parameter* $W_*$. *Let* $\sigma$ *be L-Lipschitz and be bounded by B (either of these can be taken as infinity if desired). Lastly, let* $s = \min\{\text{std}(h), \text{std}(h')\}$. *If* $\|h - h'\|_2 \leq \epsilon$, *then*

$$\begin{aligned}
\|y(h) - y'(h')\|_2 \leq & \epsilon + \sqrt{n} \|W_d\|_2 \|W_u\|_2 \min\{2B, L(\|W_g\|_2 \epsilon/s + \sqrt{n}\delta)\} \\
& + \|W_d\|_2 (\|W_u\|_2 \epsilon/s + \sqrt{n}\delta) \min\{B, \|\sigma(W'_g \tilde{h}')\|\} \\
& + \sqrt{n} \|W'_u\|_2 \delta \min\{B, \|\sigma(W'_g \tilde{h}')\|_2\}
\end{aligned}$$

*Proof.* Let $\tilde{h} = \text{LayerNorm}(h)$ and $\tilde{h}' = \text{LayerNorm}(h')$. Thus,

$$\begin{aligned}
\|y(h) - y'(h')\|_2 \leq & \|h - h'\|_2 + \|W_d(\sigma(W_g \tilde{h}) \odot W_u \tilde{h}) - W'_d(\sigma(W'_g \tilde{h}') \odot W'_u \tilde{h}')\|_2 \\
\leq & \epsilon + \|W_d\|_2 \|\sigma(W_g \tilde{h}) \odot W_u \tilde{h} - \sigma(W'_g \tilde{h}') \odot W'_u \tilde{h}'\|_2 \\
& + \|W_d - W'_d\|_2 \|\sigma(W'_g \tilde{h}') \odot W'_u \tilde{h}'\|_2 \\
\leq & \epsilon + \|W_d\|_2 \|\sigma(W_g \tilde{h}) \odot W_u \tilde{h} - \sigma(W'_g \tilde{h}') \odot W_u \tilde{h}\|_2 \\
& + \|W_d\|_2 \|\sigma(W'_g \tilde{h}') \odot W_u \tilde{h} - \sigma(W'_g \tilde{h}') \odot W'_u \tilde{h}'\|_2 \\
& + \|W_d - W'_d\|_2 \|\sigma(W'_g \tilde{h}') \odot W'_u \tilde{h}'\|_2
\end{aligned}$$

Since $\sigma$ is bounded by $B$,

$$
\begin{aligned}
\|y(h) - y'(h')\|_2 \leq & \epsilon + \|W_d\|_2 \min\{2B\|W_u\tilde{h}\|_2, \|\sigma(W_g\tilde{h}) \odot W_u\tilde{h} - \sigma(W_g'\tilde{h}') \odot W_u\tilde{h}\|_2\} \\
& + \|W_d\|_2 \min\{B\|W_u\tilde{h} - W_u'\tilde{h}'\|_2, \|\sigma(W_g'\tilde{h}') \odot W_u\tilde{h} - \sigma(W_g'\tilde{h}') \odot W_u'\tilde{h}'\|_2\} \\
& + \|W_d - W_d'\|_2 \min\{B\|W_u'\tilde{h}'\|_2, \|\sigma(W_g'\tilde{h}') \odot W_u'\tilde{h}'\|_2\} \\
\leq & \epsilon + \|W_d\|_2 \min\{2B, \|\sigma(W_g\tilde{h}) - \sigma(W_g'\tilde{h}')\|_2\}\|W_u\tilde{h}\|_2 \\
& + \|W_d\|_2 \min\{B, \|\sigma(W_g'\tilde{h}')\|\}\|W_u\tilde{h} - W_u'\tilde{h}'\|_2 \\
& + \|W_d - W_d'\|_2 \min\{B, \|\sigma(W_g'\tilde{h}')\|_2\}\|W_u'\tilde{h}'\|_2
\end{aligned}
$$

Also, since $\sigma$ is $L$-Lipschitz,

$$
\begin{aligned}
\|y(h) - y'(h')\|_2 \leq & \epsilon + \|W_d\|_2 \min\{2B, L\|W_g\tilde{h} - W_g'\tilde{h}'\|_2\}\|W_u\tilde{h}\|_2 \\
& + \|W_d\|_2 \min\{B, \|\sigma(W_g'\tilde{h}')\|\}\|W_u\tilde{h} - W_u'\tilde{h}'\|_2 \\
& + \|W_d - W_d'\|_2 \min\{B, \|\sigma(W_g'\tilde{h}')\|_2\}\|W_u'\tilde{h}'\|_2
\end{aligned}
$$

From Lemma E.2,

$$
\begin{aligned}
\|y(h) - y'(h')\|_2 \leq & \epsilon + \|W_d\|_2 \min\{2B, L(\|W_g\|_2\|\tilde{h} - \tilde{h}'\|_2 + \sqrt{n}\delta)\}\|W_u\|_2\|\tilde{h}\|_2 \\
& + \|W_d\|_2 \min\{B, \|\sigma(W_g'\tilde{h}')\|\}(\|W_u\|_2\|\tilde{h} - \tilde{h}'\|_2 + \sqrt{n}\delta) \\
& + \|W_d - W_d'\|_2 \min\{B, \|\sigma(W_g'\tilde{h}')\|_2\}\|W_u'\|_2\|\tilde{h}'\|_2
\end{aligned}
$$

Finally, using Lemma E.1,

$$
\begin{aligned}
\|y(h) - y'(h')\|_2 \leq & \epsilon + \|W_d\|_2 \min\{2B, L(\|W_g\|_2\epsilon/s + \sqrt{n}\delta)\}\|W_u\|_2\sqrt{n} \\
& + \|W_d\|_2 \min\{B, \|\sigma(W_g'\tilde{h}')\|\}(\|W_u\|_2\epsilon/s + \sqrt{n}\delta) \\
& + \delta \min\{B, \|\sigma(W_g'\tilde{h}')\|_2\}\|W_u'\|_2\sqrt{n}
\end{aligned}
$$

$\square$

**Corollary E.4.** *For $\sigma$ being the sigmoid function, Lemma E.3 becomes*

$$
\begin{aligned}
\|y(h) - y'(h')\|_2 \leq & \epsilon + \frac{\|W_d\|_2\|W_u\|_2\epsilon}{s}\left(\frac{\sqrt{n}}{4}\|W_g\|_2 + 1\right) \\
& + \sqrt{n}\delta\left(\frac{\sqrt{n}}{4}\|W_d\|_2\|W_u\|_2 + \|W_d\|_2 + \sqrt{n}\|W_u'\|_2\right)
\end{aligned}
$$

*Proof.* Sigmoids have Lipschitz constant 1/4 and is bounded by 1, and

$$
\begin{aligned}
\|y(h) - y'(h')\|_2 \leq & \epsilon + \frac{\sqrt{n}}{4}\|W_d\|_2\|W_u\|_2(\|W_g\|_2\epsilon/s + \sqrt{n}\delta) \\
& + \|W_d\|_2(\|W_u\|_2\epsilon/s + \sqrt{n}\delta) \\
& + \sqrt{n}\|W_u'\|_2\delta \\
= & \epsilon + \frac{\|W_d\|_2\|W_u\|_2\epsilon}{s}\left(\frac{\sqrt{n}}{4}\|W_g\|_2 + 1\right) \\
& + \sqrt{n}\delta\left(\frac{\sqrt{n}}{4}\|W_d\|_2\|W_u\|_2 + \|W_d\|_2 + \sqrt{n}\|W_u'\|_2\right)
\end{aligned}
$$

$\square$

This provides an understanding of the error that occurs in the GLU layers of the transformer. Next, the error through the attention layers is investigated.

**Lemma E.5.** *Consider the following function:*

$$y(H) = \mathrm{Attn}(H) := H + W_v \mathrm{LayerNorm}(H)\mathrm{softmax}(\mathrm{LayerNorm}(H)^\top W_q^\top W_k \mathrm{LayerNorm}(H))$$

*Assume that for each location $i \in [w]$, it holds that $\|H_i - H'_i\|_2$. Under the assumption that $\|W_* - W'_*\|_2 \leq \delta$ for every parameter $W_*$. Also, let $s = \min\{\mathrm{std}(H_w), \mathrm{std}(H'_w)\}$. Then*

$$\begin{aligned}
\|y(H)_w - y'(H')_w\|_2 \leq &\epsilon + \|W_v\|_2\sqrt{n}\|a - a'\|_1 \\
&+ \|W_v\|_2\epsilon/s \\
&+ \delta\sqrt{n}
\end{aligned}$$

*Proof.* Let $\tilde{H} = \mathrm{LayerNorm}(H)$ and $\tilde{H}' = \mathrm{LayerNorm}(H')$, where $\mathrm{LayerNorm}(\cdot)$ is applied elementwise. Further, let $v = W_v\tilde{H}$, $q = W_q\tilde{H}$, $k = W_k\tilde{H}$, and $a_i = \mathrm{softmax}(q_w^\top k_i/\sqrt{n})$. Then

$$\begin{aligned}
\|y(H)_w - y'(H')_w\|_2 \leq &\|H_w - H'_w\|_2 + \|W_v\tilde{H}\|_2\|a - a'\|_2 \\
&+ \|W_v\tilde{H} - W'_v\tilde{H}'\|_2\|a'\|_2 \\
\leq &\|H_w - H'_w\|_2 + \|W_v\tilde{H}\|_2\|a - a'\|_2 \\
&+ \|W_v\|_2\|\tilde{H} - \tilde{H}'\|_2\|a'\|_2 \\
&+ \|W_v - W'_v\|_2\|\tilde{H}'\|_2\|a'\|_2
\end{aligned}$$

Note that since $\sum_i a_i = 1$, $\|a_i\|_2 \leq 1$, so the bound can be updated as

$$\begin{aligned}
\|y(H)_w - y'(H')_w\|_2 \leq &\epsilon + \|W_v\|_2\frac{mn\epsilon}{2s}(\|W_q\|_2 + \|W'_q\|_2)\|W_k\|_2 + mn^{3/2}\|W_v\|_2\delta(\|W_k\|_2 + \|W'_q\|_2) \\
&+ \|W_v\|_2\frac{\sqrt{m}\epsilon}{s} + \delta\sqrt{mn}
\end{aligned}$$

since if $\|\tilde{H}_i - \tilde{H}'_i\|_2 \leq \epsilon/s$, then $\|\tilde{H} - \tilde{H}'\|_2 \leq \sqrt{m}\epsilon/s$. What remains to be seen is how $\|a - a'\|_2$ behaves. The first thing to note is that the softmax operator is $1/2$-contractive, so

$$\begin{aligned}
\|a - a'\|_2 \leq &\frac{1}{2}\|\tilde{H}_w^\top W_q^\top W_k\tilde{H} - \tilde{H}'_w{}^\top W'_q{}^\top W'_k\tilde{H}'\|_2 \\
\leq &\frac{\sqrt{m}}{2}\max_i\{\|\tilde{H}_w - \tilde{H}'\|_2\|W_q^\top W_k\tilde{H}_i\|_2 \\
&+ \|\tilde{H}'_w\|_2\|W_q - W'_q\|_2\|W_k\tilde{H}_i\|_2 \\
&+ \|\tilde{H}'_w\|_2\|W'_q\|_2\|W_k\tilde{H}_i - W'_k\tilde{H}'_i\|\} \\
\leq &\frac{\sqrt{m}}{2}\max_i\{\frac{\epsilon}{s}\|W_q\|_2\|W_k\|_2\sqrt{n} \\
&+ \sqrt{n}\delta\|W_k\|_2\sqrt{n} \\
&+ \sqrt{n}\|W'_q\|_2(\|W_k\|\frac{\epsilon}{s} + \sqrt{n}\delta)\} \\
= &\frac{\sqrt{mn}\epsilon}{2s}(\|W_q\|_2 + \|W'_q\|_2)\|W_k\|_2 + \sqrt{m}n\delta(\|W_k\|_2 + \|W'_q\|_2)
\end{aligned}$$

$\square$

# F. Experiment Details

**Datasets.** We evaluate our approach on six benchmarks: three commonsense reasoning datasets, **BoolQ** (Clark et al., 2019), **Winogrande** (Sakaguchi et al., 2021), **ARC Challenge** (Clark et al., 2018); two mathematical reasoning, **GSM8K** (Cobbe et al., 2021) and **AQuA** (Ling et al., 2017); and one long-context task, **ListOps** from Long Range Arena (Tay et al., 2020; Nangia & Bowman, 2018). Prompt templates and dataset details is available in Appendix F.1.

**Hyperparameters and Evaluation.** For each configuration, we select the optimal learning rate via grid search over five values using a validation set. Tables 1 and 2 report the mean accuracy of **five independent runs**; the standard deviations are provided in Appendix F.3, hyperparameter search space in Appendix F.2, and training setup in Appendix F.5.

## F.1. Dataset details

Table 7 shows the number of train, test, and validation set of each dataset we use. We use the default train and test split from the huggingface repository of each dataset. When there is only 2 splits (e.g., in BoolQ and GSM8K) we split train set with ratio 80:20 and random seed 42, and take the :20 split as validation set.

*Table 7.* Dataset details. Validation is created from train; table shows original dataset sizes.

| Dataset | Train | Test | Validation |
| --- | --- | --- | --- |
| BoolQ | 9,430 | 3,270 | – |
| Winogrande | 9,250 | 1,770 | 1,270 |
| ARC-Challenge | 1,120 | 1,170 | 299 |
| GSM8K | 8,790 | 1,320 | – |
| AQuA | 97,467 | 254 | 254 |
| ListOps | 96,000 | 2,000 | 2,000 |

Huggingface links:

- BoolQ

- Winogrande subset: winogrande_debiased

- ARC-Challenge subset: ARC-Challenge

- GSM8K subset: main

- AQuA

- ListOps

Table 8 details the prompt used for each dataset.

## F.2. Hyperparameters

For all model and method combination, we keep the size of hyperparam space the same to 5. Based on the number of parameters, we shift left and right. Lesser parameters are shifted to the right (higher values), and more parameters to the left. This is to account for the learnability in different parameter count.

For 1B models, we sweep over the following learning rates:

- SFT: $[5e^{-6}, 1e^{-5}, 2e^{-5}, 5e^{-5}, 1e^{-4}]$

- LoRA: $[5e^{-5}, 1e^{-4}, 3e^{-4}, 7e^{-4}, 1.5e^{-3}]$

- Activation steering r=8 (ReFT and ours): $[5e^{-4}, 7.5e^{-4}, 1e^{-3}, 2e^{-3}, 3e^{-3}]$

*Table 8.* Dataset Prompt Templates

| Dataset | Prompt Template |
|---|---|
| **BoolQ** | Answer the question with a true/false based on the given passage.
Passage: {`passage`}
Question: {`question`} |
| **Winogrande** | Please choose the correct answer to fill in the blank to complete the given sentence: {`sentence`}
Option1: {`option1`}
Option2: {`option2`} |
| **ARC-Challenge** | Answer the following multiple-choice question.
Question: {`question`}
{`options`} |
| **GSM8K** | {`question`} |
| **AQuA** | Answer the following multiple-choice math question.
Question: {`question`}
{`options`} |
| **ListOps** | Evaluate the value of the following nested list expression.
Return only the final numeric result inside `<answer>...</answer>`.

Expression:
{`expression`} |

- Activation steering r=1 and vector: $[5e^{-4}, 1e^{-3}, 2e^{-3}, 3e^{-3}, 5e^{-3}]$

- Joint: $[1e^{-4}, 2e^{-4}, 5e^{-4}, 7e^{-4}, 1e^{-3}]$

For 4B models, we sweep over the following learning rates:

- SFT: $[2e^{-6}, 5e^{-6}, 1e^{-5}, 2e^{-5}, 3e^{-5}]$

- LoRA: $[3e^{-5}, 7e^{-5}, 1.5e^{-4}, 3e^{-4}, 7e^{-4}]$

- Activation steering r=8 (ReFT and ours): $[3e^{-4}, 5e^{-4}, 8e^{-4}, 1.2e^{-3}, 2e^{-3}]$

- Activation steering r=1 and vector: $[5e^{-4}, 7e^{-3}, 1e^{-3}, 2e^{-3}, 3e^{-3}]$

- Joint: $[1e^{-4}, 2e^{-4}, 5e^{-4}, 7e^{-4}, 1e^{-3}]$

For ReFT, we apply the intervention on all layers (as recommended by their paper), and tune the intervention location between the last prompt token (the default in the code) and (p+7, s+7), their best hyperparameter for GSM8K. For LoFIT, we intervene on all heads.

*Table 9.* Chosen hyperparameters to reproduce experiment numbers (Ours) in Table 1

| Model | BoolQ | WinoG | GSM8K | ListOps |
|---|---|---|---|---|
| Llama-3.2-1B | $1e^{-3}$ | $7.5e^{-4}$ | $7.5e^{-4}$ | $7.5e^{-4}$ |
| gemma-3-1b | $5e^{-4}$ | $7.5e^{-4}$ | $2e^{-3}$ | $1e^{-3}$ |
| Qwen 3 4B | $5e^{-4}$ | $1e^{-3}$ | $7.5e^{-4}$ | $2e^{-3}$ |
| Llama 3.1 8B | $5e^{-4}$ | $5e^{-4}$ | $5e^{-4}$ | $2e^{-4}$ |

## F.3. Performance Standard Deviation

Table 12 shows the standard deviation across the 5 runs of the numbers shown in Table 1. We report a single SFT run, hence std dev is not applicable.

*Table 10.* Chosen hyperparameters to reproduce experiment numbers (Ours) in Table 2

| Model | Variant | BoolQ | WinoG | GSM8K | ListOps |
|---|---|---|---|---|---|
| Llama-3.2-1B | Ours r=1 | $2e^{-3}$ | $5e^{-4}$ | $5e^{-3}$ | $3e^{-3}$ |
| | Ours vector | $5e^{-4}$ | $1e^{-3}$ | $2e^{-3}$ | $1e^{-3}$ |
| gemma-3-1b | Ours r=1 | $2e^{-3}$ | $5e^{-4}$ | $1e^{-3}$ | $2e^{-3}$ |
| | Ours vector | $5e^{-3}$ | $5e^{-3}$ | $1e^{-3}$ | $5e^{-3}$ |
| Qwen 3 4B | Ours r=1 | $7e^{-4}$ | $7e^{-4}$ | $1e^{-4}$ | $7e^{-4}$ |
| | Ours vector | $2e^{-3}$ | $2e^{-3}$ | $7e^{-4}$ | $1e^{-3}$ |
| Llama 3.1 8B | Ours r=1 | $7e^{-4}$ | $7e^{-4}$ | $5e^{-4}$ | $1e^{-3}$ |
| | Ours vector | $7e^{-4}$ | $7e^{-4}$ | $1e^{-3}$ | $1e^{-3}$ |

*Table 11.* Chosen hyperparameters to reproduce experiment numbers (Joint Orth) in Table 5

| Model | BoolQ | WinoG | GSM8K |
|---|---|---|---|
| Llama-3.2-1B | $7e^{-4}$ | $1e^{-4}$ | $1e^{-4}$ |
| gemma-3-1b | $5e^{-4}$ | $7.5e^{-4}$ | $2e^{-3}$ |
| Qwen 3 4B | $5e^{-4}$ | $1e^{-3}$ | $7.5e^{-4}$ |

Table 13 shows the standard deviation across the 5 runs of the numbers shown in Table 2.

Table 14 shows the standard deviation of the numbers shown in Table 5.

### F.4. Hardware Details

All experiments are conducted on a single NVIDIA A100-SXM4-40GB for 1B and 4B models, and on a single NVIDIA A100-SXM4-80GB for 8B model (Section 6.3).

### F.5. Training Setup.

All models are trained using the AdamW optimizer (Loshchilov & Hutter, 2017) with a batch size of 8 (simulated via gradient accumulation where necessary for larger models). For SFT and LoRA, we employ a cosine learning rate scheduler with a warmup ratio of 0.1 and 0.06, respectively. In contrast, following the minimalist design of activation-based steering, our adapter and the ReFT baseline do not utilize weight decay, warmup, or a learning rate scheduler. BoolQ, Winogrande, ARC Challenge, AQuA and ListOps are trained for 1 epoch and GSM8K for 3 epochs.

Our RL experiments on section 6.2 uses 512 sequence length, 1025 completion length, batch size 6, gradient accumulation 4 steps and 6 rollout generations. We sweep 12 learning rates and average across 3 random seeds. We use DeepSeek's default chat template for formatting.

### F.6. Code

Our code is available here: https://github.com/SprocketLab/steerling.git

## G. Extra Experiment Results

### G.1. Activation adapter matrix norm after orthogonality projection

*Table 12.* Standard deviation of experiments in Table 1

| Model | Method | BoolQ | WinoG | GSM8K | ListOps |
|---|---|---|---|---|---|
| Llama-3.2-1B | LoRA | $8.0 \times 10^{-1}$ | 4.8 | $5.5 \times 10^{-1}$ | $7.4 \times 10^{-1}$ |
| | ReFT | $6.8 \times 10^{-1}$ | $2.1 \times 10^{-1}$ | $3.5 \times 10^{-1}$ | $4.3 \times 10^{-1}$ |
| | Ours | $2.7 \times 10^{-1}$ | $9.2 \times 10^{-1}$ | $4.4 \times 10^{-1}$ | $2.9 \times 10^{-1}$ |
| gemma-3-1b | LoRA | $6.1 \times 10^{-1}$ | 3.3 | $3.0 \times 10^{-1}$ | 9.3 |
| | ReFT | $6.0 \times 10^{-1}$ | 0.0 | $6.5 \times 10^{-1}$ | $6.7 \times 10^{-1}$ |
| | Ours | $3.3 \times 10^{-1}$ | $7.8 \times 10^{-1}$ | $2.0 \times 10^{-1}$ | $1.7 \times 10^{-1}$ |
| Qwen 3 4B | LoRA | $3.3 \times 10^{-1}$ | $8.0 \times 10^{-1}$ | $7.2 \times 10^{-1}$ | $7.3 \times 10^{-1}$ |
| | ReFT | $3.8 \times 10^{-1}$ | $1.6 \times 10^{-1}$ | 1.20 | 1.01 |
| | Ours | $2.4 \times 10^{-1}$ | 1.1 | 1.0 | $2.5 \times 10^{-1}$ |
| Llama 3.1 8B | SFT | 0.0 | 0.0 | 0.0 | 0.0 |
| | LoRA | $4.0 \times 10^{-1}$ | $5.2 \times 10^{-1}$ | $8.2 \times 10^{-1}$ | 1.45 |
| | ReFT | $7.0 \times 10^{-1}$ | 1.8 | $6.3 \times 10^{-1}$ | $4.2 \times 10^{-1}$ |
| | Ours | $3.7 \times 10^{-1}$ | $3.3 \times 10^{-1}$ | 1.2 | 1.9 |

*Table 13.* Standard deviation of experiments in Table 2

| Model | Method | BoolQ | WinoG | GSM8K | ListOps |
|---|---|---|---|---|---|
| Llama-3.2-1B | LoFIT | $8.0 \times 10^{-1}$ | 4.8 | $5.5 \times 10^{-1}$ | $7.4 \times 10^{-1}$ |
| | JoLA | $6.8 \times 10^{-1}$ | $2.1 \times 10^{-1}$ | $3.5 \times 10^{-1}$ | $4.3 \times 10^{-1}$ |
| | Ours vector | $4.2 \times 10^{-1}$ | $5.2 \times 10^{-1}$ | $4.5 \times 10^{-1}$ | $3.4 \times 10^{-1}$ |
| | Ours r=1 | $4.7 \times 10^{-1}$ | $4.7 \times 10^{-1}$ | $5.1 \times 10^{-1}$ | $3.7 \times 10^{-1}$ |
| gemma-3-1b | LoFIT | $5.1 \times 10^{-1}$ | $7.1 \times 10^{-1}$ | 1.02 | $3.0 \times 10^{-1}$ |
| | JoLA | $5.3 \times 10^{-1}$ | $3.2 \times 10^{-1}$ | 1.10 | $2.5 \times 10^{-1}$ |
| | Ours vector | $8.2 \times 10^{-1}$ | $8.9 \times 10^{-1}$ | $6.4 \times 10^{-1}$ | $2.7 \times 10^{-1}$ |
| | Ours r=1 | 1.20 | $7.9 \times 10^{-1}$ | $8.2 \times 10^{-1}$ | $2.4 \times 10^{-1}$ |
| Qwen 3 4B | LoFIT | $5.4 \times 10^{-1}$ | $9.2 \times 10^{-1}$ | $6.2 \times 10^{-1}$ | $3.9 \times 10^{-1}$ |
| | JoLA | $2.2 \times 10^{-2}$ | 1.03 | $5.4 \times 10^{-1}$ | $1.1 \times 10^{-1}$ |
| | Ours vector | $2.8 \times 10^{-1}$ | $5.1 \times 10^{-1}$ | $8.5 \times 10^{-1}$ | $4.1 \times 10^{-1}$ |
| | Ours r=1 | $2.5 \times 10^{-1}$ | $5.8 \times 10^{-1}$ | $7.5 \times 10^{-1}$ | $4.4 \times 10^{-1}$ |
| Llama 3.1 8B | LoFIT | $6.4 \times 10^{-1}$ | 1.2 | 1.3 | $4.8 \times 10^{-1}$ |
| | JoLA | $7.6 \times 10^{-1}$ | 1.67 | $9.7 \times 10^{-1}$ | $4.2 \times 10^{-1}$ |
| | Ours vector | $1.8 \times 10^{-3}$ | $1.2 \times 10^{-3}$ | $1.3 \times 10^{-3}$ | $1.1 \times 10^{-3}$ |
| | Ours r=1 | $2.1 \times 10^{-1}$ | $4.7 \times 10^{-1}$ | 1.1 | 1.9 |

*Table 14.* Standard deviation of experiments in Table 5

| Model | Method | BoolQ | WinoG | GSM8K |
|---|---|---|---|---|
| Llama-3.2-1B | LoRA | $8.17 \times 10^{-1}$ | 4.86 | $5.50 \times 10^{-1}$ |
| | Adapter | $4.74 \times 10^{-1}$ | $4.10 \times 10^{-1}$ | $1.48 \times 10^{-1}$ |
| | Joint | $8.14 \times 10^{-1}$ | 3.03 | 1.58 |
| | Joint Orth | $5.08 \times 10^{-1}$ | 4.02 | $5.97 \times 10^{-1}$ |
| gemma-3-1b | LoRA | $6.11 \times 10^{-1}$ | 3.34 | $3.03 \times 10^{-1}$ |
| | Adapter | $6.91 \times 10^{-1}$ | $6.42 \times 10^{-1}$ | $5.46 \times 10^{-1}$ |
| | Joint | $9.28 \times 10^{-1}$ | $9.09 \times 10^{-1}$ | 2.84 |
| | Joint Orth | $9.32 \times 10^{-1}$ | 1.32 | 1.04 |
| Qwen 3 4B | LoRA | $3.29 \times 10^{-1}$ | $8.02 \times 10^{-1}$ | $7.16 \times 10^{-1}$ |
| | Adapter | $3.08 \times 10^{-1}$ | $7.23 \times 10^{-1}$ | $6.58 \times 10^{-1}$ |
| | Joint | 1.02 | $6.14 \times 10^{-1}$ | $9.86 \times 10^{-1}$ |
| | Joint Orth | $5.41 \times 10^{-1}$ | 1.34 | $1.30 \times 10^{-1}$ |

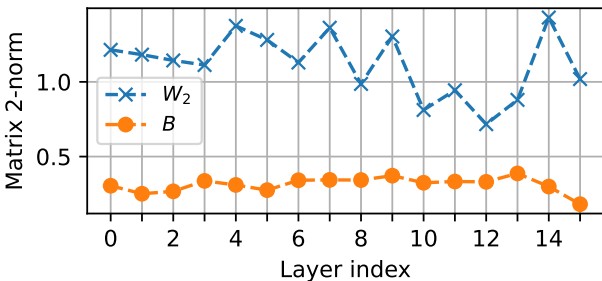

*Figure 6.* $W_2$ after orthogonality projection retains its magnitude.

