# OpenReview forum: "Weight Updates as Activation Shifts: A Principled Framework for Steering"
_ICML.cc/2026/Conference — ICML 2026 regular_

### Official Review · Reviewer_dZQu · 2026-03-07

**Soundness:** 4
**Presentation:** 3
**Significance:** 4
**Originality:** 4
**Overall Recommendation:** 5
**Confidence:** 4

**Summary:**

This paper establishes a principled theoretical framework for "Activation Steering" in Large Language Models (LLMs). The authors derive a first-order equivalence between weight-space updates and activation-space shifts using Taylor expansion. Based on this theory, the paper identifies the "post-block output" as a more expressive intervention site than "post-MLP" locations, as it captures the full residual stream. Furthermore, the authors propose Joint-Orth, an orthogonality-constrained joint adaptation method that learns in both weight and activation spaces while minimizing functional redundancy.

**Compliance With Llm Reviewing Policy:**

Affirmed.

**Key Questions For Authors:**

•	Could you elaborate on how your method performs in scenarios where significant weight updates are required, and whether your framework can be extended to incorporate higher-order terms for more robust theoretical guarantees?
•	Could you provide a more rigorous theoretical analysis showing how the skip-connection interacts with attention updates in the residual stream, and why this location uniquely enables capturing attention effects?
•	Table 3 shows that Joint-Orth doesn't always improve performance over individual methods (e.g., WinoG on Qwen). Could you elaborate on why the orthogonality constraint sometimes fails to provide benefits, and whether there are specific characteristics of certain datasets or model architectures that determine when the constraint is beneficial?

**Limitations:**

yes

**Strengths And Weaknesses:**

Strengths:
•	Technical novelty and innovation: The paper bridges the gap between empirical activation steering and formal fine-tuning theory. The use of the right singular vectors from SVD to enforce orthogonality between the steering adapter and weight updates is a well-motivated to preventing subspace collapse.
•	Experimental rigor and validation: The methodology is validated across diverse modern architectures and scales. The task selection—covering commonsense reasoning (Winogrande), mathematics (GSM8K), and long-context reasoning (ListOps), provides a multi-dimensional assessment of the framework.
•	Significance of contributions: By providing a "principled" foundation for steering, this work moves the field away from black-box heuristics toward interpretable and predictable parameter-efficient fine-tuning (PEFT).
•	Clarity: Joint-Orth is conceptually straightforward: learn two branches while discouraging them from occupying the same subspace. This is a clean design choice, and the paper communicates the intent clearly.
Weaknesses:
•	Confounded Gains: The central claim is that “post-block is more principled,” but the main setting often defaults to global intervention across layers. If baselines (e.g., ReFT/JoLA) use different layer-coverage strategies, gains may be driven by coverage rather than the locus itself. Without carefully decoupled ablations, the “post-block advantage” is not yet a causal conclusion.
•	Gradient competition under orthogonality: Early in training, if the weight-update branch and the activation-steering branch compete for similar gradient directions, the orthogonality projection may repeatedly remove the effective gradient for the activation branch. This can act like a dynamic suppression of the activation branch’s effective learning rate (effective LR shrinkage), potentially slowing convergence or causing early stagnation.
•	Degeneracy risk: If the task’s optimal adaptation direction is extremely strong and highly aligned, the activation-steering branch under repeated orthogonal projection may gradually degenerate toward (near-)zero, effectively becoming a zero matrix. In that case, Joint-Orth’s advantage over pure LoRA would vanish. The paper does not provide experiments demonstrating robustness on such “highly aligned direction” settings, nor does it show that Joint-Orth reliably outperforms single-branch tuning under strong directional alignment.
•	Add the ablations please: (i) post-MLP vs post-block under matched layer coverage and matched parameter budgets; (ii) post-block on a subset of layers vs post-block on all layers; and report variance.
•	Add system-level evaluation please: peak training memory, inference overhead of extra operators, and throughput.
•	Please add a “highly aligned direction” controlled setting (synthetic or real) where a single low-rank direction suffices, and measure whether the steering branch collapses toward zero and whether Joint-Orth maintains gains.
•	To improve readability, all symbols should be defined in the main text, not in the appendix.

---

> ### Author Rebuttal · Authors · 2026-03-30
>
> Thank you for noting the novelty and significance of our work! We answer each comment next.
>
> - **W1: Confounded gains.** All of the baselines in our experiments (ReFT, LoFIT, JoLA) intervene at all layers to match our parameter count. We have clarified this in our revised draft. Additionally, we provide an ablation <!-- in E1 of common response -->that compares pre-MLP, post-MLP, and post-block under identical parameterization (same rank and adapter architecture), isolating the effect of the intervention site. Post-block consistently outperforms both alternatives. The results are included in Q1 for reviewer zn9H.
> - **W2: Gradient under orthogonality.** Thank you for pointing this out! The orthogonality projection is applied every 100 training steps, giving both branches sufficient room to learn distinct directions between successive projections. We clarify this detail in our revised manuscript. Empirically, in Figure 6 (App. G.1) we show that the activation adapter retains significant magnitude (0.8–1.4 across layers) after training.
> - **W3 Degeneracy risk.** Indeed, such highly aligned settings are possible. In these settings, **either steering or LoRA alone suffices**, and orthogonally projecting one of these components does not hinder the model from using the other component freely. An experiment of this can be seen under W6.
> - **W4: Ablations.** For the post-block vs. other intervention sites ablation, we refer the reviewer to <!--E1 in the common response.--> Q1 in the response to reviewer zn9H. For the layer coverage ablation, we report results below. For each layer count, we sample 3 random contiguous subsets and report the mean accuracy.
>
> Model: Llama-3.1-8B, Dataset: Winogrande
>
> |Num layers|Mean Acc
> |-|-|
> |1|57.8|
> |4|60.8|
> |8|62.7|
> |16|68.3|
> |All (32)|87.2|
>
> Standard deviation across layers is 4.4, and accuracy increases monotonically with layer count, supporting our global scope all-layer intervention (Sec 5.1).
>
> - **W5: System-level eval.** Our adapter adds two matrix-vector products ($W_1 \in \mathbb{R}^{r \times d}$, $W_2 \in \mathbb{R}^{d \times r}$) per layer, resulting in inference overhead of $2 × r × d$ FLOPs per layer.
>
> |Model|d|Adapter FLOPs/layer|Layer FLOPs (MLP+Attn)|Overhead|
> |-|-|-|-|-|
> |Llama-3.2-1B|2048|32K|~67M|0.048%|
> |Llama-3.1-8B|4096|65K|~243M|0.027%|
>
> For training memory, the following are GPU memory allocations for training Llama-3.2-1B with batch size 8.
>
> |Method|Peak GPU Memory (GB)|
> |-|-|
> |SFT|26.0|
> |LoRA (r=8)|15.0|
> |Ours (r=8)|8.4|
>
>
>
> - **W6: Controlled setting.**
>   We use BoolQ on Llama-3.2-1B as a naturally "highly aligned" setting, where a rank-1 weight update is largely sufficient.
>
> |Method|Accuracy|
> |-|-|
> |LoRA r=1|87.0|
> |Joint-Orth (r=1)|87.3|
>
> Performance is maintained. Unlike the higher-rank setting in Figure 6 of our manuscript — where $W_2$ retains magnitude 0.8–1.4 across layers — we observe here ([link](https://anonymous.4open.science/r/actvweight_icml26_rebuttal-C022/ablation_norm.pdf)) that the orthogonality projection reduces $W_2$ to near-zero. This demonstrates that, when a single weight-space direction suffices, the activation branch defers rather than competing. When additional capacity is beneficial (as in a higher-rank setting), both branches retain magnitude and contribute complementary features.
>
> - **W7: Readability.** Thank you for this suggestion! We moved symbol definitions to the main paper.
> - **Q1: Significant weight update and higher order terms.**
>     We refer the reviewer to <!--common response E2--> the response to Q3 for reviewer zn9H, where we show that under settings requiring substantial behavioral changes like instruction tuning and RL, our approach maintains strong performance.
>     We note that for pre-trained models, the loss landscape typically acts roughly linear locally to the pre-trained model weights. Second-order effects are likely small in the regimes of interest; this is why we did not pursue a higher-order analysis.
> - **Q2: Theoretical analysis.**
>     For a concrete example of steering uniquely capturing the attention outputs, the intuitive example in Section 4 shows how steering includes a term proportional to $x$ in the second component, which is missing for fine-tuning. This $x$ is the input to the GLU: the sum of the block input and the attention output.
> - **Q3: Joint-Orth limitation.**
>     Indeed, adding orthogonality does not always improve performance. We find this behavior especially interesting; our hypothesis for why this takes place is related to an experimental choice for training efficiency: enforce orthogonality every 100 steps rather than every iteration. It is possible that the steering component is learning something meaningful, but is projected away only every 100 steps, allowing for it to relearn only for it to be projected again.

---

> > ### Author Rebuttal · Reviewer_dZQu · 2026-04-06
> >
> > Thank you very much for the explanation, but l'm keeping my score.

---

### Official Review · Reviewer_xWD8 · 2026-03-07

**Soundness:** 3
**Presentation:** 3
**Significance:** 3
**Originality:** 3
**Overall Recommendation:** 5
**Confidence:** 3

**Summary:**

In this paper, the fine-tuning of LLMs is analyzed and compared to steering. The analysis is used to design a more parameter-efficient fine-tuning approach that is based on post-block (rather than pre-MLP or post-MLP) steering. The proposed algorithm is reported to empirically perform well.

**Compliance With Llm Reviewing Policy:**

Affirmed.

**Final Justification:**

The rebuttal fully addressed the concerns raised in this review. I have increased the confidence score accordingly.

**Key Questions For Authors:**

1) Could the explanations in the Sections 3.2 and 3.3 be extended to address the lack of clarity specified in the Weaknesses section of this review?

**Limitations:**

Yes

**Strengths And Weaknesses:**

# Strengths
- The paper in a formal and principled way examines the connection between fine-tuning and steering
- The gained insights are empirically tested by informing the design of a new algorithm
- Experimental design is solid, with experiments including multiple models and having been executed several times, with standard deviation reported

# Weaknesses
- While the paper is written in good English and has a comprehensible structure, the logic within each section often escaped this reviewer. See concrete examples below.
- Section 3.2 could provide more detailed explanation of how exactly post-MLP steering contributes the missing term.
- In Section 3.3, the explanation of why the adapter interactions can be ignored is not clear to this reviewer.
- Theorem 3.1 is not properly set up and explained. The statement comes out of the blue and the conclusions drawn from it do not clearly follow (for this reviewer, who has not encountered principal angles before).

---

> ### Author Rebuttal · Authors · 2026-03-30
>
> We thank the reviewer for their support and helpful comments! These were extremely useful for adding clarity to our draft.
>
> **Section 3.2 Comments.** We added the following to understand post-MLP steering covering the $W_d$ term. The main insight is that post-MLP steering can directly add an output-space correction analogous to the missing $(\Delta W_d)m$ term, without requiring that correction to pass through the GLU
>
> **Section 3.3 Comments.**  Theorem 3.1 elaborates on the conditions where the interaction between the MLP output and the full block output is negligible. Specifically, Theorem 3.1 says that for any post-MLP steering, the optimal post-block steering to match the post-MLP steering is determined by how the geometry of the post-block location (with its interaction terms) differs from the post-MLP location.
>
> **Theorem 3.1 Discussion.** We agree that the exposition about principal angles/vectors is light. To address this, in our updated draft, we replaced the second paragraph after Theorem 3.1 with the following:
> "Principal angles measure the difference between two subspaces, where the $k$-th principal angle is the smallest angle between the two in directions orthogonal to the previous $k-1$ principal angles. The theorem above indicates the relevant quantity relating post-MLP steering $A_p$ to post-block steering $A$ is the similarity in the subspaces spanned by the kernel matrix of the post-MLP $X$ and the kernel matrix of the post-block activations $X+Y$. These are identical to the subspaces spanned by their right singular vectors $V$ and $V'$. It is important to note that this is measuring the kernel matrix rather than the covariance matrix, indicating a similarity needs to exist in *sample-space rather than feature-space*. What matters is that the geometry of the activations does not change much between before and after the MLP. If the MLP distorts the sample-to-sample geometry greatly, changing the kernel matrix, the resulting $\sin \theta_i$ will increase, decreasing the possible quality between post-MLP and post-block steering."
>
> **Theorem 3.1 Setup.** We have changed the paragraph prior to the theorem to say:
> "With the similarity between steering and fine-tuning understood, we investigate how closely a linear post-block steering update can mimic a ReFT-style post-MLP steering. These ReFT-style steerings will be matrices multiplying the associated component of the base model (either the output of just the MLP or the entire block)."

---

> > ### Author Rebuttal · Reviewer_xWD8 · 2026-04-01
> >
> > Thank you for the response, the explanations and modifications make the paper much clearer to this reviewer.

---

### Official Review · Reviewer_QaDn · 2026-03-12

**Soundness:** 2
**Presentation:** 2
**Significance:** 2
**Originality:** 2
**Overall Recommendation:** 3
**Confidence:** 3

**Summary:**

The paper investigates the relationship between weight-space fine-tuning and activation-space steering in large language models.
It proposes a first-order equivalence mapping between these two adaptation spaces. Based on this analysis, the authors argue that "post-block" activation steering—intervening after the skip connection is added back to the MLP output—is more expressive than pre- or post-MLP interventions because it captures the full residual stream. Additionally, the paper introduces a "joint adaptation" method that trains both weight and activation parameters simultaneously, utilizing an orthogonality constraint to prevent the two updates from collapsing into redundant subspaces.

**Compliance With Llm Reviewing Policy:**

Affirmed.

**Final Justification:**

The paper addresses an interesting problem, but the current version lacks clarity in several parts. The authors have outlined revisions that would likely improve the presentation significantly. However, since the revised manuscript is not available, I base my evaluation on the current version and keep my original score. I slightly lower my confidence to account for the expected improvements and to support a possible acceptance.

**Key Questions For Authors:**

- Can you thoroughly revise Sections 3 and 4 to unify the notation across the manuscript and provide step-by-step derivations for your approximations, rather than skipping steps? Clarifying this is necessary to evaluate the soundness of your core claims.
- For additional questions see Weaknesses

**Limitations:**

Yes.

**Strengths And Weaknesses:**

**Strengths**

**Strong empirical results**: The empirical performance of the post-block steering approach is solid. The method manages to achieve accuracy within 0.2%-0.9% of full-parameter fine-tuning on average across the tested tasks while training only 0.04% of the parameters. It also demonstrates competitive performance against established baselines like LoRA and ReFT.

**Practical joint training trick**: The intuition to enforce an orthogonality constraint between the output spaces of the steering intervention and the weight update is a clever and practical engineering solution. Figure 5 provides good empirical validation that this prevents functional redundancy.

**Weaknesses**

**Presentation and Writing**: The paper is poorly written and lacks a coherent structure. The narrative is difficult to follow, particularly in the theoretical sections due to the lack of notation, which obscures the core claims. Moreover equations are not numbered, making it difficult to reference them.

**Originality and Significance**: The overall contribution feels minor and incremental. While the authors frame their approach as a "principled framework," moving an adapter to the "post-block" position is ultimately a structural heuristic.

**Weak and Unclear Theoretical Analysis**: The theoretical foundation is weak, relying on approximations that are difficult to track, making it nearly impossible to follow. Specifically, several parts of the paper are exceptionally hard to understand:

- **Section 3.2**: the papar jumps from the definition of the GLU layer to the final derivation, using notation never introduced (a_*, m) resulting in a section impossible to comprehend

- **Section 3.3**: The setup for this theorem is incredibly dense. The sudden introduction of matrices X and Y with the definition "Y_i = h_i + Attn(h_i)" is poorly motivated prior to the theorem statement. Furthermore, the subsequent explanation regarding "right principal angles rather than the left ones" mapping to sample space instead of feature space is confusing and lacks mathematical rigor.

- **Section 3.3**: it's not clear why post-block is better than post-mlp, the autors says it's due to 40%-70% of the output carried by the residual connection, but it's not clear why this should justify the architectural choice.

- **Section 4**: The definition of g^(x) arbitrarily mixes variables x, F(x), delta h, and delta W in a way that feels completely disjointed from the notation established in Section 3. The "intuitive example" provided fails to clarify the mechanics and instead adds to the confusion due to its high level of abstraction.

The choice of Post-Block locus over post-MLP and pre-MLP is not supported by experimental results.

---

> ### Author Rebuttal · Authors · 2026-03-30
>
> Thank you for the careful read and the excellent suggestions, in particular on notation---we believe it has made the paper substantially easier to read. We clarified some of our notational choices and discussion in the revised draft. The following is a breakdown of these modifications:
>
> - **Section 3.2 Comments.** Thank you for pointing this out! The definitions of $a_*$ and $m$ were present early in the appendix, and not in the main body. These definitions have been added to Section 3.2 in the revised draft.
> - **Section 3.3 Comments.** We have changed $X$ and $Y$ to $Y^{pa}$ and $Y^{mlp}$ for the post-attention output and the MLP output for clarity. The motivation behind using the present shorthand of $X$ and $Y$ was to contract the size of the formula in the theorem; using full notation makes it unreadably large.
> - **Singular Vector Comments.** Thank you for noting this! Indeed, the discussion of singular vectors may be difficult to follow for those not closely acquainted with these tools. We added some exposition surrounding this in the revised draft; the specific additions are described in detail in the response to reviewer xWD8. To briefly summarize the exposition, by definition, left principal vectors come from the singular vectors of the covariance matrix, while right principal vectors come from the singular vectors of the kernel matrix. Hence, right singular vectors (as is the definition of $V$ and $V'$) represent properties of how samples relate to each other rather than the spread of the data.
> - **Section 3.3 Comments.** Post-block steering multiplies both the output of the GLU as well as the residual connection. Any adaptation that needs to be done to the attention outputs for a post-MLP steer must deal with the non-linearities of the GLU, which can zero out important directions. Steering post-block can directly multiply the residual connection. The corruption in the signal by adding $GLU(h + Attn(h))$ to $h + Attn(h)$ is established in Theorem 3.1.
> - **Section 4 Comments.** We have replaced the inputs from $x$ to $h$, which we agree adds to the clarity. We note that **none of the rest of the notation conflicts with previous sections**. The notation $F$ is for convenience, simplifying the internals of the GLU. The choice of notation for the output of a steered and finetuned model was not necessary until that proposition, where we used $\hat g$. Hence, it does not conflict with any previous section.
> For the details of the differences between steering and fine-tuning, $g$ contains a fine-tuning update ($W_2 \rightarrow W_2 + \delta W_2$, as is standard) and a ReFT-style post-block steering (a post multiplication by a matrix $I + \delta h$).

---

> > ### Author Rebuttal · Reviewer_QaDn · 2026-04-02
> >
> > I thank the authors for their effort and for addressing the reviewers’ comments.
> >
> > My concerns regarding writing quality and clarity, however, remain. I would need to carefully read the fully revised version of the paper to properly assess whether these aspects have improved. Additionally, I find the issue of novelty, as raised by another reviewer, to be relevant.
> >
> > Therefore, the main reasons behind my score are the limited technical and theoretical contributions: a) issues with clarity and notation, and b) the number of assumptions and approximations required. That said, I still consider this to be a solid piece of work.

---

> > > ### Author Response · Authors · 2026-04-03
> > >
> > > We thank the reviewer for their continued engagement with our work! While we cannot upload the revised version of the manuscript due to conference policy, we are happy to provide additional portions of the revised version with, for example, notation clarified. We will highlight some of the key changes, reflecting improvements to our manuscript. We will additionally discuss novelty and assumptions and the relevant changes we have made.
> > >
> > > **Section 3.2.**
> > > We have updated the text to the following:
> > >
> > > "...a small change to the activations $h$ and a small change to the parameters $W_d$, $W_g$, $W_u$; with the form
> > > $\Delta GLU_{steer}(h) = W_d\Big[\big(\phi'(a_g)\odot a_u\big)\odot (W_g \Delta h)+\phi(a_g)\odot (W_u \Delta h)\Big]+O(\|\Delta h\|^2)$
> > > and
> > > $\Delta GLU_{FT}(h) =(\Delta W_d)m + W_d\Big[\big(\phi'(a_g)\odot a_u\big)\odot ((\Delta W_g) h)+\phi(a_g)\odot ((\Delta W_u) h)
> > > \Big] +O((\|\Delta W_d\| + \|\Delta W_g\| + \|\Delta W_u\|)^2),$
> > > where
> > > $a_g = W_g h,\qquad
> > > a_u = W_u h,\qquad
> > > m = \phi(a_g)\odot a_u, \text{ and }
> > > y = W_d m.$
> > > These follow directly from a first-order expansion of the GLU equation $GLU(h) = W_d(\phi(W_g h) \odot W_u h)$. ..."
> > >
> > > **Derivation of the former.**
> > > To derive the first-order approximation for $GLU_{steer}$, we take a differential of the output:
> > > $\partial GLU_{steer}(h) = W_d\left[\partial (\phi(W_g h) \odot W_u h)\right] = W_d\left[\partial (\phi(W_g h)) \odot W_u h +  \phi(W_g h) \odot \partial (W_u h)\right] = W_d\left[\phi'(a_g) \odot a_u \odot W_g (\partial h) +  \phi(a_g) \odot W_u (\partial h)\right].$
> > > The first-order approximation follows from the first-order Taylor expansion of $GLU_{steer} (h).$
> > >
> > > **Section 3.3 -  Theorem 3.1 Setup.** We have changed the paragraph prior to the theorem to add motivation, saying:
> > > "With the similarity between steering and fine-tuning understood, we investigate how closely a linear post-block steering update can mimic a ReFT-style post-MLP steering. These ReFT-style steerings will be matrices multiplying the associated component of the base model (either the output of just the MLP or the entire block)."
> > >
> > > **Section 4.**
> > > We have modified the text before Proposition 4.1 to the following:
> > > "The first core difference between steering and fine-tuning is given by the following proposition. First, to get a handle on the notation, we condense the MLP into $W_2 F(h)$. Here, $W_2$ is the down-projection for the MLP, $h$ is the input to the MLP, and $F$ encapsulates the rest of the MLP machinery (note that the parameterization of $F$ will be different for different choices of MLP, such as a GLU). The result after the residual connection is written as $\hat{g}(h) = h + W_2 F(h)$. Post-block steering will act as a matrix product on the left by $I + \delta h$ in the ReFT style (identity + low rank on the residual) and fine-tuning will replace $W_2$ with $W_2 + \delta W_2$."
> > >
> > > **On novelty.** We agree that the specific adapter parameterization is not the core novelty of our work. Instead, **the novelty lies in the analytical framework that replaces heuristic-driven steering design with principled guidance**. In addition, we introduce the notion of orthogonally-constrained joint training, which has not been explored by prior work. We have revised the text to clarify this in our Introduction contribution paragraph.
> > >
> > > **On assumptions.** We agree that our analysis primarily requires  the approximation of both steering and fine-tuning being local/first-order. Our intent is to show that there are similarities in the terms that each can express---thus revealing what each intervention point can or cannot do---rather than a direct equivalence between steering and fine-tuning. Our empirical results support the usefulness of the theory, despite being first-order, indicating that post-block steering does result in the highest performance of all loci. Additionally, the joint orthogonality results are more general, holding without this first order assumption. We have modified our contribution paragraph in Introduction accordingly.

---

### Official Review · Reviewer_zn9H · 2026-03-15

**Soundness:** 2
**Presentation:** 2
**Significance:** 3
**Originality:** 2
**Overall Recommendation:** 4
**Confidence:** 4

**Summary:**

This paper studies parameter-efficient adaptation via hidden-state interventions rather than direct weight updates. It provides a first-order analysis relating activation perturbations to weight updates in transformer MLP blocks, using this to justify post-block steering over pre-MLP and post-MLP alternatives. The method is a post-block bottleneck adapter applied across layers. The paper also argues steering and weight updates are complementary, proposing joint training of activation adapters with LoRA under an orthogonality constraint. Experiments compare against SFT, LoRA, ReFT, LoFiT, and JoLA on BoolQ, Winogrande, GSM8K, and ListOps across several model families, reporting gains over prior steering baselines and performance close to SFT at much smaller parameter budgets.

**Compliance With Llm Reviewing Policy:**

Affirmed.

**Final Justification:**

The rebuttal addressed my main concerns. I have updated my score.

**Key Questions For Authors:**

1. Can you provide a clean ablation of pre-MLP vs. post-MLP vs. post-block under the same adapter parameterization, rank, and layer-coverage policy? This would directly test the main mechanistic claim and would improve my assessment if the advantage of post-block remains clear.

2. Can you provide parameter-matched baselines, especially low-budget LoRA and controls at the same total budget as Joint-Orth? Without this, it is difficult to tell whether the gains come from the proposed orthogonal decomposition or simply from extra trainable capacity.

3. Have you tested the method on generation-heavy tasks where the intervention must remain effective over multiple decoding steps rather than mostly short-answer settings? Positive evidence there would strengthen the practical significance claim.

**Limitations:**

Yes, the limitations section is reasonably candid about the first-order/local nature of the theory and about not fully disentangling locus from intervention policy, which I appreciate.

**Strengths And Weaknesses:**

Strengths:
- The paper is clearly motivated and generally well written. The overall narrative from theory to method to experiments is easy to follow. Proposition 4.1 is the most conceptually interesting result in the paper: it explains cleanly why steering and fine-tuning can play different functional roles because of the skip connection, which provides a plausible motivation for joint training. The empirical section is also reasonably broad for a PEFT paper, with several model families, several tasks, multiple strong baselines, and appendix-level details on hyperparameters, variance, and training setup.

- The paper also makes a useful practical point that intervention location matters. The reported comparisons to ReFT, LoFiT, and JoLA suggest that the proposed post-block configuration is a strong activation-steering baseline, and the paper does a decent job of tying those empirical results back to its theoretical discussion.

Weaknesses:
- My main concern is that the novelty is more limited than the paper suggests. Relative to ReFT, this is still fundamentally a learned low-parameter representation intervention; the main differences are the post-block placement, the all-layer application policy, and the theoretical justification. More broadly, the method is close to adapter-style methods cited in the paper, so the main novelty is the argument for this locus rather than a fundamentally new adaptation mechanism. I would like the paper to position itself more carefully on this point.

- A second issue is that the theoretical equivalence claims rely on fairly strong assumptions. The central connection between steering and weight updates is explicitly local and first-order, so it should be viewed as a linearized approximation rather than a practical equivalence theorem for fine-tuning dynamics. Some of the geometric conditions used in the post-block discussion are also stated somewhat vaguely and are not empirically checked. Parts of the theory therefore feel more like a clean formalization of an intuitive point, namely that later interventions can express effects earlier interventions cannot, than a genuinely surprising discovery. Theorem 3.1 is technically fine, but it more directly establishes post-block > post-MLP in expressivity than post-block ≈ SFT.

- I also do not think the empirical evidence fully isolates the paper’s main causal claim. The gains combine intervention locus, all-layer application, and the specific adapter design. The paper itself notes in the limitations that it does not fully disentangle where to intervene from how to intervene, but that is an important gap because the central contribution is precisely about identifying the right intervention site. Similarly, the “global scope” claim feels under-justified: the theory is largely layer-local, and I did not find a strong argument that it implies interventions should be applied at every layer rather than some subset.

- Some of the mechanistic evidence is also weaker than the prose suggests. Figure 2 uses the ratio of MLP output norm to block output norm, but that does not directly tell us how much of the fine-tuning-induced change lies in the MLP pathway versus the residual/attention pathway. Likewise, the paper does not measure the decomposition that would most directly support the discussion around the output-projection term, namely how much of fine-tuning’s effect comes from output-projection changes versus gate/up-projection changes.

- I also found parts of the framing overstated. The claim that “activation steering serves as the next step in this evolution” implies more of a hierarchy than the results support; the empirical story here is still a Pareto tradeoff with SFT, not replacement of SFT. Similarly, the “completes the triangle” language overstates the connection to the ICL/fine-tuning literature, since those works study implicit learning dynamics during inference, whereas this paper studies explicit trained adapters under first-order approximations. These are related, but not equivalent in the strong sense the metaphor suggests.

Overall, I think the paper contains a real idea and some useful analysis, but the combination of limited novelty relative to closely related methods, strong first-order assumptions, and incomplete empirical isolation keeps me below the acceptance bar.

---

> ### Author Rebuttal · Authors · 2026-03-30
>
> Thank you for noting the clear motivation and actionable insights from our work! We respond to the reviewer's comments:
>
> - **W1: Novelty.** **Our core contribution is the analytical framework** that replaces heuristic-driven design with principled guidance, also leading to orthogonally-constrained updates (Prop 4.1). Indeed, the adapter parameterization is not our primary contribution, and we have adjusted the paper's positioning to make this clearer.
>
> - **W2: Our theoretical contributions.** Indeed, the core of our approach is motivated by first-order approximation. As the parameter regime of interest near a set of pre-trained weights is typically close to linear, we hypothesize that first-order approximation is sufficient, and we empirically validate this in Sec 6.2.
>
> - **W3: Global scope.** Our focus is on establishing the equivalence between fine-tuning and steering, so **the model we attempt to match fine-tunes modules at every layer**, and to match each of these updates, **we steer at every layer**. This is clarified in our revised draft. **Our baselines (ReFT, LoFIT, JoLA) are set to intervene at all layers** to match. Additional empirical evidence for steering at all layers can be found in Q4 for reviewer dZQu.
>
> - **W4: Mechanistic evidence.** Thank you for this suggestion! We provide two new measurements:
>
> 1. **Fine-tuning-induced shift decomposition (new figure, [[link1](https://anonymous.4open.science/r/actvweight_icml26_rebuttal-C022/rebuttal_delta_out%20(1).png)]):** We plot $\frac{\|\Delta y_{\text{MLP}}\|}{\|\Delta y_{\text{MLP}}\| + \|\Delta y_{\text{Attn}}\|}$, where $\Delta y_{\text{module}} = \text{module}_{FT}(h) - \text{module}_{base}(h)$, using the same setup as Figure 2. MLP-induced shifts account for at most ~70% of the total change at any layer, providing more evidence that post-MLP steering misses a large portion of the fine-tuning-induced change in the attention pathway.
>
> 2. **Weight update decomposition across MLP submodules (new figure, [[link2](https://anonymous.4open.science/r/actvweight_icml26_rebuttal-C022/rebuttal_mlp_abs.png)]):** We plot $\|\Delta W_u\|_F$, $\|\Delta W_g\|_F$, and $\|\Delta W_d\|_F$ across layers. All three matrices receive updates of comparable size, indicating the down-projection term **constitutes a non-negligible portion of the total MLP update**.
>
> - **W5: Framing.** We agree and have fine-tuned the framing in our updated draft. We now (1) position activation steering as a complementary, cost-efficient alternative to SFT rather than its successor, and (2) describe our connection to the ICL/fine-tuning literature as an adjacent analytical bridge rather than a completion of the triangle.
>
> - **Q1: Clean ablation.** We ablate pre-MLP, post-MLP, and post-block *identically* using rank-8 linear adapters applied at all layers, isolating the effects of intervention site from different adapter architectures.
>
> Llama-3.2-1B
> |Method|BoolQ|WinoG|GSM8K|ListOps
> |-|-|-|-|-|
> Pre-MLP|85.0|51.5|31.0|64.1
> Post-MLP|84.8|51.0|28.5|64.4
> Post-block|**86.2**|**60.1**|**31.5**|**65.4**
>
> Llama-3.1-8B
> |Method|BoolQ|WinoG|GSM8K|ListOps
> |-|-|-|-|-|
> Pre-MLP|91.4|86.3|39.6|**72.3**
> Post-MLP|92.2|87.0|41.6|68.6
> Post-block|**92.3**|**87.2**|**43.4**|69.1
>
> **Post-block outperforms pre- and post-MLP in 7/8 settings**. The exception is ListOps on 8B, where pre-MLP leads and post-block outperforms post-MLP. These results **support that intervention site matters, and post-block is the strongest default choice**.
> - **Q2: Joint-Orth LoRA baseline.** We include LoRA (r=16) as a baseline with *more* parameters than Joint-Orth.
>
> |Model|Dataset|LoRA r16 (0.9%)|Joint-Orth (0.79%)|
> |-|-|-|-|
> |Llama-3.2-1B|GSM8K|31.2|31.1|
> ||BoolQ|88.4|**88.5**|
> ||WinoG|**64.3**|59.3|
> |Gemma-3-1b|GSM8K|23.8|**24.4**|
> ||BoolQ|**84.8**|**84.8**|
> ||WinoG|**51.8**|50.9|
> |Qwen 3 4B|GSM8K|38.3|**38.8**|
> ||BoolQ|91.0|**91.4**|
> ||WinoG|**84.6**|83.5|
>
> **Joint-Orth matches or outperforms LoRA r=16 on most settings despite using fewer parameters**. On Winogrande, where Joint-Orth underperforms LoRA r=16, it generally improves over naive Joint training confirming the value of the orthogonality constraint.
>
> - **Q3: Generation Heavy Task.** We evaluate on two settings requiring substantial behavioral changes and multi-step generation:
>
> 1. **Instruction tuning** (Llama-3.1-8B, AlpacaEval): Open-ended generation, testing whether the intervention remains effective across extended decoding.
>
> |Method|Params|LC Win Rate|
> |-|-|-|
> |Full SFT|100%|11.49|
> |Ours (r16)|0.05%|11.34|
> |LoRA (r16)|0.52%|9.59|
> |ReFT (r16)|0.05%|10.96|
>
> 2. **RL fine-tuning** (DeepSeek-R1-Distill-Qwen-1.5B, GSM8K): Generation involves multi-step chain-of-thought reasoning.
>
> |Method|Params (%)|Pass@1|
> |-|-|-|
> |Base model|-|10.2|
> |LoRA|0.52%|81.5±0.7|
> |Ours (nonlinear)|0.04%|**84.7±0.5**|
> |Ours (linear)|0.04%|84.3±0.8|
>
> These results suggest that **while our framework is first-order, the design choices it derives transfer to settings with larger weight updates.**

---

> > ### Author Rebuttal · Reviewer_zn9H · 2026-04-05
> >
> > Thank you for the rebuttal, I will adjust my score.

---

### Decision · Program_Chairs · 2026-04-30

**Decision:**

Accept (regular)

**Comment:**

**Summary:** This paper establishes a first-order equivalence between weight-space updates and activation-space interventions in transformers, using this to argue that post-block steering is a theoretically grounded and expressive intervention site. It further proposes joint adaptation (Joint-Orth), which trains activation adapters alongside LoRA with an orthogonality constraint to prevent redundancy. Experiments across multiple models and tasks show post-block steering achieves accuracy within 0.4%-1.5% of full fine-tuning at 0.04% of parameters.

**Strengths:** The analytical framework connecting steering and fine-tuning is well-motivated and yields actionable design guidance. Strong empirical results across multiple architectures and tasks, including generation-heavy settings added during rebuttal. The orthogonality-constrained joint training is a clean and practical idea with good empirical validation.

**Weaknesses:** The theoretical analysis relies on first-order approximations and the expressivity results (post-block > post-MLP) are somewhat expected given the skip connection structure. Writing quality, particularly notation and exposition in Sections 3 and 4, was flagged by multiple reviewers. The novelty relative to existing adapter methods could be more carefully positioned.

**Recommendation:** Weak accept. The authors should improve presentation clarity in the final version.